# Pairwise Maximum Likelihood for Multi-Class Logistic Regression Model with Multiple Rare Classes

Xuetong Li [1]   Danyang Huang [2][3]   Hansheng Wang [4]

## Abstract

We study in this work the problem of multi-class logistic regression with one major class and multiple rare classes, which is motivated by a real application in TikTok live stream data. The model is inspired by the two-class logistic regression model of Wang (2020) but with surprising theoretical findings, which in turn motivate new estimation methods with excellent statistical and computational efficiency. Specifically, since rigorous theoretical analysis suggests that the resulting maximum likelihood estimators of different rare classes should be asymptotically independent, we consider to solve multiple pairwise two-class logistic regression problems instead of optimizing the joint log-likelihood function with computational challenge in multi-class problem, which are computationally much easier and can be conducted in a fully parallel way. To further reduce the computation cost, a subsample-based pairwise likelihood estimator is developed by downsampling the major class. We show rigorously that the resulting estimators could be as asymptotically efficient as the global maximum likelihood estimator under appropriate regularity conditions. Extensive simulation studies are presented to support our theoretical findings and a TikTok live stream dataset is analyzed for illustration purpose.

## 1. Introduction

We study in this work the problem of multi-class logistic regression with one major class and multiple rare classes.

[1]School of Mathematics and Statistics, Xi'an Jiaotong University, Xi'an, China [2]Center for Applied Statistics, Renmin University of China, Beijing, China [3]School of Statistics, Renmin University of China, Beijing, China [4]Guanghua School of Management, Peking University, Beijing, China. Correspondence to: Danyang Huang <dyhuang89@126.com>.

*Proceedings of the 42$^{nd}$ International Conference on Machine Learning*, Vancouver, Canada. PMLR 267, 2025. Copyright 2025 by the author(s).

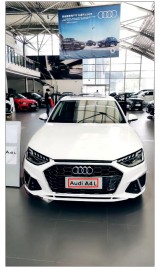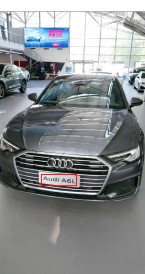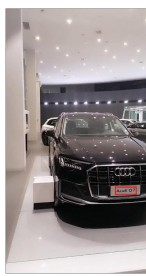

*Figure 1.* Examples of Audi TikTok live streams. Each red bounding box annotates a local region containing a car plate, which contains valuable model identity information.

This problem is well motivated by many real-world classification tasks, such as recognizing rare animal species (George et al., 2023; Mou et al., 2023) and detecting uncommon objects in traffic scenes (Mullapudi et al., 2021a; Zhang et al., 2024). In these real-world settings, the categories of interest are often extremely rare. As pointed out by Mullapudi et al. (2021a;b), it is typically easy to collect a large number of negative instances in this case, which can be obtained through weak- or semi-supervised methods (Ratner et al., 2017; Chen et al., 2020), or as a by-product of multi-label annotation (Deng et al., 2014), or with random sampling. This leads to an imbalanced image classification problem, where the "background" category overwhelmingly dominates, and rare classes (e.g., target objects) represent the minority.

To fix the idea, consider for example a car plate recognition problem in TikTok live streams. Figure 1 shows screenshots from TikTok live streams sponsored by different Audi car dealers. Through the live stream, the car dealers wish to entice as many potential buyers as possible from the TikTok users to make purchasing decisions (Su et al., 2023). To this end, it is important to understand TikTok users' possible purchase intention, which can be partially reflected in their viewing behaviors (Ma, 2021; Sun et al., 2024). Interestingly, this important information is accurately shown on the car plate; see the red bounding boxes in Figure 1. Then, how to statistically analyze the graphical information provided by the car plate so that the car model can be accurately and automatically recognized becomes the key problem. Un-

fortunately, the position of the car plate in the image is not known in advance and its position is quite random. To capture the car plate, a large number of bounding boxes are randomly generated over the entire screenshot image. This leads to a large number of sub-image samples; see Figure 4(c). Specifically, since these screenshot images are high-resolution images with small car plates, one can expect that most sub-images would reflect the background content. They can be labeled as $k = 0$ ("background"), which is the major class and accounts for more than 95% of all samples. The remaining sub-images (less than 5%) are about the car plates and are classified into $K = 8$ classes indexed by $1 \leq k \leq K$; see Figure 5 for an intuitive understanding. We formulate this problem as a classical multi-class logistic regression problem with a total of $(K + 1) = 9$ classes. To accomplish this task, a multi-class logistic regression model or its sophisticated variants can be used.

Most existing literature focuses on the multi-class logistic regression model (Nelder & Wedderburn, 1972) under a classical situation with broad applications (Maalouf, 2011; Tanha et al., 2020). By a classical situation, we mean that the number of classes is not very big, the class distribution is fairly balanced and the feature dimension is relatively low. Unfortunately, this is not the case for many real applications (e.g., the TikTok live streams problem as discussed before) (Nelder & Wedderburn, 1972). Instead, practitioners often face challenging problems with a number of classes (Abramovich et al., 2021), multiple rare classes (Mullapudi et al., 2021a), and high dimensional features (Wang et al., 2023). This makes the associated parameter estimation, e.g., the maximum likelihood estimator (MLE), extremely difficult. For example, if the traditional Newton-Raphson (NR) algorithm is to be used for computing the MLE, a Hessian matrix (i.e., the second–order derivative of the log-likelihood function) with dimension $(p + 1)K \times (p + 1)K$ needs to be inverted (McCullagh & Nelder, 2019). This makes the computation cost extremely high. As a useful alternative, various gradient based methods have been developed and popularly used with better computational feasibility. However, they suffer from the tuning parameter (i.e., the learning rate) selection problem and slow convergence rate issue (Zhu et al., 2021). All those challenges inspire us to search for a new method, which can compute the MLE in a more efficient way.

To solve the problem, we first consider the standard MLE, of which the asymptotic theory has been well understood in the past literature, but for the situation with fairly balanced class distribution (Fox, 2015; McCullagh & Nelder, 2019). We are then motivated to fill this important theoretical gap. Our asymptotic theory suggests that, for a multi-class logistic regression model with one major class and multiple rare classes, the asymptotic covariance matrix of the resulting MLE is block-diagonal. This suggests that the regression

coefficients of each rare class might be estimated separately instead of jointly, without sacrificing the asymptotic efficiency. This interesting finding inspires the idea of the pairwise likelihood estimation. Specifically, we first decompose the original $(K + 1)$–class classification problem into a total of $K$ major–and–rare class pairs. Next, the MLE of the major–and–rare class pair can be easily computed by a standard Newton-Raphson (NR) algorithm. Theoretically, we find that the resulting pairwise maximum likelihood estimator (PMLE) is statistically efficient asymptotically. Practically, the PMLE enjoys nice computational properties. First, the Hessian matrix associated with each major–and–rare class pair is only of $(p+1) \times (p+1)$–dimension, which is much smaller than $(p + 1)K \times (p + 1)K$–dimension of the full-class maximum likelihood problem. Second, the PMLE for different class pairs can be computed in a fully parallel or distributed manner.

Although the PMLE method is suitable for parallel or distributed estimation, we find that the computation cost remains significant. The main reason is that the major class is fully involved for each major–and–rare class pair. Additionally, we theoretically verify that the convergence rate of the PMLE is mainly determined by the sample size of rare classes, instead of the major class. This further implies that the sample size of the major class might be too large to be necessary for PMLE. Therefore, we are inspired to consider a much reduced sample size for the major class, so that the computation cost can be further reduced. The subsample-based pairwise likelihood estimator (SPMLE) is then constructed. Theoretically, we find that the SPMLE is also asymptotically efficient. However, compared with the global maximum likelihood estimator (GMLE) and PMLE, the SPMLE is easier to compute with much less computational cost. Extensive simulation studies are presented in this paper to demonstrate the finite sample performance of these estimators. The aforementioned TikTok Screenshots dataset is also used for illustration purpose.

The rest of this paper is organized as follows. In Section 2, we first introduce the model setting and the rare class phenomenon. We then present three estimation methods (i.e., GMLE, PMLE and SPMLE) and study their asymptotic properties. Simulation studies are given in Section 3. A Tik-Tok live stream dataset is analyzed for illustration purpose in Section 4. The article concludes with a brief discussion and concluding remarks in Section 5. All technical details are delegated to the appendix in the supplementary material.

## 2. The Asymptotic Theory

### 2.1. Model Setup and Rare Class Phenomenon

Let $(Y_i, X_i)$ be the observation collected from the $i$–th subject with $1 \leq i \leq N$. Here $Y_i \in \{0, 1, \cdots, K\}$ is the $i$–th

class label and $X_i = (X_{i1}, \cdots, X_{ip})^\top \in \mathbb{R}^p$ is the associated $p$–dimensional feature. Assume that the random vector $X_i$ is sub-gaussian in the sense that the one-dimensional marginals $a^\top X_i$ are sub-gaussian random variables for all $a \in \mathbb{R}^p$ (Vershynin, 2018). To model their regression relationship, a standard multi-class logistic model is assumed as follows (McCullagh & Nelder, 2019)

$$P(Y_i = k \mid X_i) = \frac{\exp\left(X_i^\top \beta_k + \alpha_k\right)}{1 + \sum_{k=1}^K \exp\left(X_i^\top \beta_k + \alpha_k\right)}. \quad (1)$$

Here $\beta_k = (\beta_{k1}, \cdots, \beta_{kp}) \in \mathbb{R}^p$ is the regression coefficient associated with the $k$–th rare class, and $\alpha_k \in \mathbb{R}$ is the intercept parameter accordingly. We know immediately that $P(Y_i = 0 \mid X_i) = 1 - \sum_{k=1}^K P(Y_i = k \mid X_i) = 1/\{1 + \sum_{k=1}^K \exp(X_i^\top \beta_k + \alpha_k)\}$. Accordingly, a log-likelihood function is constructed as $\mathcal{L}(\Theta) = \sum_{i=1}^N \sum_{k=0}^K a_{ik} \log w_{ik}$, where $\theta_k = (\alpha_k, \beta_k^\top)^\top \in \mathbb{R}^{p+1}$ and $\Theta = (\theta_1^\top, \cdots, \theta_K^\top)^\top \in \mathbb{R}^q$ with $q = (p+1)K$. Here $a_{ik} = I(Y_i = k) \in \{0, 1\}$ is a binary indicator and $w_{ik} = P(Y_i = k | X_i) = \exp(X_i^\top \beta_k + \alpha_k)/\{1 + \sum_{k=1}^K \exp(X_i^\top \beta_k + \alpha_k)\}$ is the response probability of the $k$–th rare class (i.e., $1 \le k \le K$). Consequently, the MLE can be obtained as $\widehat{\Theta} = \text{argmax}_\Theta \mathcal{L}(\Theta)$, where $\widehat{\Theta} = (\widehat{\theta}_1^\top, \cdots, \widehat{\theta}_K^\top)^\top \in \mathbb{R}^q$ and $\widehat{\theta}_k = (\widehat{\alpha}_k, \widehat{\beta}_k^\top)^\top \in \mathbb{R}^{p+1}$.

Recall there are a total of $(K + 1)$ classes where the 0–th class is the major class and the other $K$ classes are rare classes. To reflect this rare class phenomenon, two conditions need to be imposed. First, we require that $P(Y_i = k) \to 0$ as $N \to \infty$ for every $k$ with $1 \le k \le K$ (Wang, 2020; Li et al., 2024). Following Wang (2020), we assume that $\beta_k$ $(1 \le k \le K)$ is a fixed parameter and does not change by the sample size $N$. Otherwise, we can hardly model the regression relationship between $Y_i$ and $X_i$ in a relative stable way. Accordingly, we have to assume $\alpha_k \to -\infty$ as $N \to \infty$. To reflect the fact that $\alpha_k$ diverges with sample size $N$, we rewrite $\alpha_k$ as $\alpha_{Nk}$. Second, we expect that the number of instances belonging to different rare classes should diverge to infinity. Otherwise, the sample sizes for the rare classes might be too small to support a valid asymptotic study. Define $N_k = \sum_{i=1}^N I(Y_i = k)$ to be the size of the $k$–th class, where $I(\cdot)$ is the indicator function. We shall have $E(N_k) \approx NP(Y_i = k) \to \infty$ as $N \to \infty$ for $1 \le k \le K$. This implies that $\alpha_{Nk} + \log N \to \infty$ when $N \to \infty$. To summarize, to appropriately reflect the rare class phenomenon, two conditions must be imposed for $\alpha_{Nk}$. They are, respectively, (1) $\alpha_{Nk} \to -\infty$ and (2) $\alpha_{Nk} + \log N \to \infty$ as $N \to \infty$ for every $1 \le k \le K$.

## 2.2. Re-Parameterized Likelihood Function

We next study the theoretical implications of those two conditions in more depth. We start with the first condition $P(Y_i = k) \to 0$ as $N \to \infty$. Note that the rare class

probability can be approximated as $P(Y_i = k \mid X_i) \approx \exp(X_i^\top \beta_k + \alpha_{Nk})$. We require the sizes of different rare classes to be comparable. This means that the class sizes should not be dramatically different. Otherwise, there must exist an even rarer class among rare classes. The size of this even rarer class would be too tiny to be of any practical relevance and can hardly be studied theoretically. Accordingly, we are inspired to assume $P(Y_i = k_1 \mid X_i)/P(Y_i = k_2 \mid X_i) \approx \exp\{X_i^\top (\beta_{k_1} - \beta_{k_2})\} \exp(\alpha_{Nk_1} - \alpha_{Nk_2}) = O_p(1)$ for any $1 \le k_1, k_2 \le K$. This condition can be easily satisfied if $\alpha_{Nk_1} - \alpha_{Nk_2} = c_{k_1 k_2}$ for some fixed constant $c_{k_1 k_2}$. Equivalently, we can also assume that $\alpha_{Nk} = \alpha_N + \beta_{k0}$ for some $\alpha_N$ and $\beta_{k0}$. Accordingly, we require that $\beta_{k0}$ is a fixed constant, but (1) $\alpha_N \to -\infty$ and (2) $\alpha_N + \log N \to \infty$ as $N \to \infty$. These conditions are fairly standard in the literature. See, for example, equation (2) in Wang (2020), Section 2 in Wang et al. (2021) and Song & Zou (2024).

Unfortunately, the above nicely defined parameters $\alpha_N$ and $\beta_{k0}$ are not immediately identifiable. To see this, let $c \in \mathbb{R}$ be an arbitrary scalar. We can then redefine $\alpha_N := \alpha_N + c$ and $\beta_{k0} := \beta_{k0} - c$. One can verify that the above model setup remains valid. It would be desirable to find a practically convenient way to make both $\alpha_N$ and $\beta_{k0}$ identifiable. To this end, note that $P(Y_i \ne 0) \approx \sum_{k>0} E\{\exp(X_i^\top \beta_k + \alpha_N + \beta_{k0})\} = e^{\alpha_N} \sum_{k>0} E\{\exp(X_i^\top \beta_k + \beta_{k0})\}$. Define $c = \log\left[\sum_{k \ne 0} E\{\exp(X_i^\top \beta_k + \beta_{k0})\}\right]$. Redefine $\alpha_N := \alpha_N + c$ and $\beta_{k0} := \beta_{k0} - c$. We then have $P(Y_i \ne 0) \approx e^{\alpha_N}$. This suggests that we might define $\alpha_N = \log P(Y_i \ne 0)$. This might be a convenient way to specify $\alpha_N$. Once $\alpha_N$ is fixed, other parameters (i.e., $\beta_{k0}$ and $\beta_k$s) can be uniquely identified. Accordingly, we should have $\beta_{k0} = \alpha_{Nk} - \alpha_N = \alpha_{Nk} - \log P(Y_i \ne 0)$.

Next, define an expanded feature vector $Z_i = (1, X_i^\top)^\top \in \mathbb{R}^{p+1}$ with the intercept term included. Define $\theta_k^* = (\beta_{k0}, \beta_k^\top)^\top \in \mathbb{R}^{p+1}$ to be the associated regression coefficient parameter. Note that $\theta_k^*$ should be carefully differentiated from $\theta_k$. Recall $\theta_k = (\alpha_{Nk}, \beta_k^\top)^\top \in \mathbb{R}^{p+1}$. Thus, the only difference between $\theta_k^*$ and $\theta_k$ is the intercept term. Next, the model (1) can be re-parameterized as

$$P(Y_i = k \mid X_i) = \frac{\exp\left(Z_i^\top \theta_k^* + \alpha_N\right)}{1 + \sum_{k=1}^K \exp\left(Z_i^\top \theta_k^* + \alpha_N\right)} \quad (2)$$

for $1 \le k \le K$. Note that the model (2) is a special case of the model (1), which is a more general model. Compared with (1), the key feature of (2) lies in its assurance that different rare classes maintain comparable sizes. Next, recall that $\alpha_N = \log P(Y_i \ne 0)$. Therefore, the re-parameterized log-likelihood function can be defined as $\mathcal{L}(\Theta^*) = \sum_{i=1}^N \sum_{k=0}^K a_{ik} \log w_{ik}^*$ according to (2), where $\Theta^* = (\theta_1^{*\top}, \cdots, \theta_K^{*\top})^\top \in \mathbb{R}^q$ with $q = (p+1)K$. Recall that $a_{ik} = I(Y_i = k) \in \{0, 1\}$ is a binary indicator. However, $w_{ik}^* = P(Y_i = k | X_i) = \exp(Z_i^\top \theta_k^* +$

$\alpha_N)/\{1 + \sum_{k=1}^{K} \exp(Z_i^\top \theta_k^* + \alpha_N)\}$ is the response probability of the $k$–th rare class under the re-parameterized model (2). Accordingly, another MLE can be defined as $\widetilde{\Theta} = \text{argmax}_{\Theta^*} \mathcal{L}(\Theta^*)$, where $\widetilde{\Theta} = (\widetilde{\theta}_1^\top, \cdots, \widetilde{\theta}_K^\top)^\top \in \mathbb{R}^q$ with $\widetilde{\theta}_k^\top = (\widetilde{\beta}_{k0}, \widetilde{\beta}_k^\top)^\top$.

## 2.3. The Global Maximum Likelihood Estimation

Note that there are two GMLEs defined. The first one is $\widehat{\Theta}$, which is obtained under the model (1). The other one is $\widetilde{\Theta}$, which is derived with the help of the re-parameterized model (2). Then it is of interest to study their relationships. One can verify that the model (2) is essentially the same as the model (1). The only difference is that $\alpha_{Nk}$ in the model (1) is re-parameterized in the model (2) as $\alpha_{Nk} = \alpha_N + \beta_{k0}$. Consequently, the following relationship holds: (1) $\beta_k$ is the same for $\widetilde{\Theta}$ and $\widehat{\Theta}$; and (2) $\alpha_{Nk} = \log P(Y_i \neq 0) + \beta_{k0}$ for every $1 \leq k \leq K$. Therefore, $\widehat{\Theta} - \Theta$ and $\widetilde{\Theta} - \Theta^*$ should share the same asymptotic distribution. We then have the following theorem.

**Theorem 2.1.** *Assume that $\alpha_N \to -\infty$ and $\alpha_N + \log N \to \infty$ as $N \to \infty$. We then have $\sqrt{Ne^{\alpha_N}}(\widehat{\Theta} - \Theta) = \sqrt{Ne^{\alpha_N}}(\widetilde{\Theta} - \Theta^*) \to_d N(0, \Sigma^{*-1})$ with $\Sigma^* = diag\{E\{\exp(Z_i^\top \theta_k^*)Z_i Z_i^\top\} : 1 \leq k \leq K\}$ as $N \to \infty$.*

The technical proof of Theorem 2.1 is given in the supplementary material Appendix A.1. By Theorem 2.1, we know that $\widehat{\Theta}$ (or equivalently $\widetilde{\Theta}$) is $\sqrt{Ne^{\alpha_N}}$–consistent and asymptotically normal. The asymptotic covariance matrix is given by $\Sigma^{*-1} = diag\{\Sigma_k^{*-1} : 1 \leq k \leq K\}$ and $\Sigma_k^* = E\{\exp(Z_i^\top \theta_k^*)Z_i Z_i^\top\}$. This implies two interesting findings. First, the convergence rate of $\widetilde{\Theta}$ is $\sqrt{Ne^{\alpha_N}}$ instead of $\sqrt{N}$. This indicates that, with multiple rare classes, the convergence rate is mainly determined by the sample size of the rare classes instead of the total sample size. Similar theoretical findings were also obtained by Wang (2020) but for an imbalanced two-class problem. Second, we surprisingly find that $\Sigma^{*-1} \in \mathbb{R}^{q \times q}$ is a matrix of a block diagonal structure. It contains a total of $K$ blocks with the $k$–th block given by $\Sigma_k^{*-1} \in \mathbb{R}^{(p+1) \times (p+1)}$. By this asymptotic covariance matrix, we know that different $\widetilde{\theta}_k$s are asymptotically independent with each other. Furthermore, one can verify that the computational cost of the GMLE is $O(Np^2K^2 + p^3K^3)$ if a standard NR algorithm is used. This implies that the standard NR algorithm can hardly be used for the case with both high feature dimension $p$ and multiple rare classes $K$.

## 2.4. The Pairwise Maximum Likelihood Estimation

The surprising findings about asymptotic independence in Theorem 2.1 suggest that the regression coefficients associated with different rare classes might be estimated separately instead of jointly. Accordingly, we need to convert the orig-

inal $(K + 1)$–class problem into a total of $K$ two-class problems. Each two-class problem corresponds to the major class and one rare class. This leads to a major–and–rare class pair; see Figure 2. In other words, we take the major class as the benchmark class and one other rare class (i.e., $k$) as the matching class.

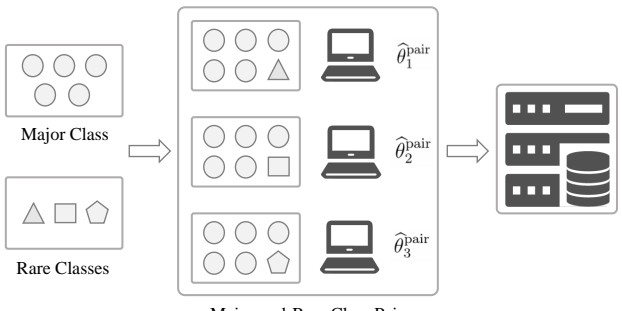

Figure 2. Illustration of the parallel or distributed computation for the PMLE. The whole dataset contains four classes with the circle class as the major class. This leads to three major–and–rare class pairs. By solving three two-class logistic regression problems, the PMLE can be obtained.

By the model (2) and condition on this major–and–rare class pair, we obtain a two-class logistic regression model as

$$P\Big(Y_i = k \mid Z_i, Y_i \in \{0, k\}\Big) = \frac{\exp(Z_i^\top \theta_k^* + \alpha_N)}{1 + \exp(Z_i^\top \theta_k^* + \alpha_N)}$$

for every rare class $1 \leq k \leq K$. This suggests a pairwise log-likelihood function as $\mathcal{L}_{0k}(\theta_k^*) = \sum_{i=1}^{N} a_{ik}(Z_i^\top \theta_k^* + \alpha_N) - (a_{ik} + a_{i0}) \log\{1 + \exp(Z_i^\top \theta_k^* + \alpha_N)\}$. The PMLE can be easily obtained as $\widehat{\theta}_k^{\text{pair}} = \text{argmax}_{\theta_k^*} \mathcal{L}_{0k}(\theta_k^*)$, whose theoretical properties are given by Theorem 2.2 as follows.

**Theorem 2.2.** *Assume that $\alpha_N \to -\infty$ and $\alpha_N + \log N \to \infty$ as $N \to \infty$. Then for $1 \leq k \leq K$, we have $\sqrt{Ne^{\alpha_N}}(\widehat{\theta}_k^{\text{pair}} - \theta_k^*) \to_d N(0, \Sigma_k^{*-1})$ with $\Sigma_k^* = E\{\exp(Z_i^\top \theta_k^*)Z_i Z_i^\top\}$ as $N \to \infty$.*

The proof of Theorem 2.2 is given in the supplementary material Appendix A.2. By Theorem 2.2, we know that the PMLE $\widehat{\theta}_k^{\text{pair}}$ is also $\sqrt{Ne^{\alpha_N}}$–consistent and asymptotically normal. We surprisingly find that the asymptotic covariance matrix is given by $\Sigma_k^{*-1} = \left[E\{\exp(Z_i^\top \theta_k^*)Z_i Z_i^\top\}\right]^{-1}$, which is the same as that of $\widehat{\theta}_k^{\text{mle}}$. However, the difference is that the PMLE $\widehat{\theta}_k^{\text{pair}}$ is computationally much easier since $\widehat{\theta}_k^{\text{pair}}$ can be computed in a fully parallel way for different $k$s; see Figure 2. If a standard NR algorithm is used, it could be verified that the computational cost for the PMLE is $O(N_0 p^2 K + p^3 K)$, which is much less than that of the GMLE. This makes it computationally efficient and feasible even with a large number of classes (i.e., $K$).

## 2.5. Subsample-Based Pairwise Maximum Likelihood Estimation

Compared with the GMLE, the PMLE is considerably easier to compute for two reasons. First, the dimension of Hessian matrix used in NR algorithm greatly reduces from $(p+1)K \times (p+1)K$ to $(p+1) \times (p+1)$. Second, the PMLEs for different class pairs can be computed in a fully parallel way. This makes the pairwise maximum likelihood method particularly suitable for parallel or distributed estimation. However, we find that the computation cost of this new method remains significant. This is mainly because the major class is fully involved for each major–and–rare class pair, whose sample size is about $O(N)$ order. This leads to a significant amount of unnecessary computation cost. To see this, note that the sample size of this major class is about $O(N)$ order. However by Theorem 2.2, we find that the convergence rate of $\widehat{\theta}_k^{\mathrm{pair}}$ is only of the order $1/\sqrt{Ne^{\alpha_N}}$, instead of $1/N$. Recall that the convergence rate of $\widehat{\theta}_k^{\mathrm{pair}}$ is mainly determined by the sample size of rare classes, instead of the major class. This further implies that the sample size of the major class might be too large to be necessary for pairwise maximum likelihood estimation. Thus, we are motivated to consider a much reduced sample size for the major class so that the computation cost can be further reduced.

To fix this idea, consider a fixed $k$ with $1 \leq k \leq K$. Next, for each observation $i$, define an independently and identically distributed binary random variable $b_i$ indicating whether this observation is sampled. Write $P(b_i = 1) = \pi_N$ for some sampling probability $0 < \pi_N < 1$. We next consider the following interesting conditional probability $P(Y_i = k | Z_i, Y_i = k \text{ or } Y_i = 0 \,\&\, b_i = 1)$. Note that the conditional probability $P(Y_i = k | Z_i, Y_i \in \{0, k\})$ should be evaluated whenever the $i$–th observation is generated by the rare class (i.e., $Y_i = k$) or the major class (i.e., $Y_i = 0$). However, the conditional probability $P(Y_i = k | Z_i, Y_i = k \text{ or } Y_i = 0 \,\&\, b_i = 1)$ might not be computed for an observation $i$ from the major class (i.e., $Y_i = 0$), unless it is also sampled (i.e., $b_i = 1$). Simple algebra shows that for $1 \leq k \leq K$, we have

$$P\Big(Y_i = k \mid Z_i, Y_i = k \text{ or } Y_i = 0 \,\&\, b_i = 1\Big)$$
$$= \frac{\exp(Z_i^\top \theta_k^* + \alpha_N^{\mathrm{sub}})}{1 + \exp(Z_i^\top \theta_k^* + \alpha_N^{\mathrm{sub}})},$$

where $\alpha_N^{\mathrm{sub}} = \alpha_N - \log \pi_N$. This leads to the following subsample-based pairwise likelihood function as $\mathcal{L}_{0k}^{\mathrm{sub}}(\theta_k^*) = \sum_{i=1}^N a_{ik}(Z_i^\top \theta_k^* + \alpha_N^{\mathrm{sub}}) - (a_{ik} + a_{i0}b_i) \log \{1 + \exp(Z_i^\top \theta_k^* + \alpha_N^{\mathrm{sub}})\}$ for every rare class $1 \leq k \leq K$. Then the SPMLE can be easily obtained as $\widehat{\theta}_k^{\mathrm{sub}} = \operatorname{argmax}_{\theta_k^*} \mathcal{L}_{0k}^{\mathrm{sub}}(\theta_k^*)$, whose theoretical properties are given by the following Theorem 2.3.

**Theorem 2.3.** *Assume that $\alpha_N \to -\infty$ and $\alpha_N + \log N \to \infty$ as $N \to \infty$. Further assume $\pi_N/e^{\alpha_N} \to \infty$ as $N \to \infty$. Then for $1 \leq k \leq K$, we have $\sqrt{Ne^{\alpha_N}}(\widehat{\theta}_k^{\mathrm{sub}} - \theta_k^*) \to_d N(0, \Sigma_k^{*-1})$ with $\Sigma_k^* = E\{\exp(Z_i^\top \theta_k^*)Z_i Z_i^\top\}$ as $N \to \infty$.*

The proof of Theorem 2.3 is given in the supplementary material Appendix A.3. By Theorem 2.3, we know that the SPMLE $\widehat{\theta}_k^{\mathrm{sub}}$ is also $\sqrt{Ne^{\alpha_N}}$–consistent and asymptotically normal. We find that the asymptotic covariance matrix is given by $\Sigma_k^{*-1} = \left[E\{\exp(Z_i^\top \theta_k^*)Z_i Z_i^\top\}\right]^{-1}$, which is the same as that of $\widehat{\theta}_k^{\mathrm{mle}}$ and $\widehat{\theta}_k^{\mathrm{pair}}$. This suggests that $\widehat{\theta}_k^{\mathrm{sub}}$ shares the same asymptotic efficiency with $\widehat{\theta}_k^{\mathrm{mle}}$ and $\widehat{\theta}_k^{\mathrm{pair}}$ as long as $\pi_N/e^{\alpha_N} \to \infty$ as $N \to \infty$, recalling that $\pi_N$ is the sampling probability. In this case, the sample size of the subsampled major class remains substantially larger than that of the rare class. However, it is much smaller than that of the whole major class. In this regard, the computational cost for SPMLE is $O(N_0 \pi_N p^2 K + p^3 K)$ if a standard NR algorithm is used. Since $N_0 \pi_N$ is of the order $o(N)$, this leads to a significant reduction in computation cost as compared with PMLE.

## 2.6. Some Extensions

CASE 1. In the previous subsection, the number of classes $K$ is treated as fixed. We next study the case with a diverging $K$. Specifically, the expected percentage of rare classes in this case should be even smaller. To ensure a diverging sample size for each rare class, the technical assumption $\alpha_N \to -\infty$ should be replaced by $\alpha_N + \log K \to -\infty$. The resulting theoretical behavior of our proposed methods remains fairly the same. More specifically, the convergence rate of the PMLE remains $\sqrt{Ne^{\alpha_N}}$. The PMLEs associated with different rare classes remain mutually independent asymptotically. More importantly, the resulting PMLE remains asymptotically as efficient as GMLE.

CASE 2. In the previous subsection, there exists only one major class (corresponding to $k = 0$). In fact, our method can be readily extended to a more general setting with multiple major classes. To fix the idea, consider, for example, the case with two major classes. Denote the two major classes by $k = 0$ and $k = 1$, respectively. Then, the logistic regression model becomes

$$w_{i0}^* = \frac{1}{1 + \exp\left(Z_i^\top \theta_1^*\right) + \sum_{k=2}^K \exp\left(Z_i^\top \theta_k^* + \alpha_N\right)},$$

$$w_{i1}^* = \frac{\exp\left(Z_i^\top \theta_1^*\right)}{1 + \exp\left(Z_i^\top \theta_1^*\right) + \sum_{k=2}^K \exp\left(Z_i^\top \theta_k^* + \alpha_N\right)}, \quad (3)$$

$$w_{ik}^* = \frac{\exp\left(Z_i^\top \theta_k^* + \alpha_N\right)}{1 + \exp\left(Z_i^\top \theta_1^*\right) + \sum_{k=2}^K \exp\left(Z_i^\top \theta_k^* + \alpha_N\right)} \quad (4)$$

for $2 \leq k \leq K$. The key difference is that there are two major classes involved in (3). In contrast, there is only one

major class involved in (2). To estimate the model parameters, the pairwise log-likelihood in Section 2.4 remains valid. The only difference is that the convergence rate of $\hat{\beta}_1$ becomes $\sqrt{N}$. However, the convergence rate of the parameters associated with the rare classes (i.e., $\hat{\beta}_k$ for $2 \leq k \leq K$) remains $\sqrt{Ne^{\alpha_N}}$.

## 3. Simulation Study

### 3.1. Comparing GMLE and PMLE

To demonstrate the finite sample performance of the proposed methods, a number of simulation studies are conducted. A multi-class logistic regression model (1) with multiple rare classes is used to generate the data. The covariate $X_i \in \mathbb{R}^p$ is independently generated from $N(0, \Sigma)$ with $\Sigma = (\sigma_{ij}) \in \mathbb{R}^{p \times p}$ and $\sigma_{ij} = 0.5^{|i-j|}$. The total sample sizes are set as $N = 1 \times 10^5, 2 \times 10^5$ and $5 \times 10^5$. For a fixed $N$, we set $\alpha_N = \gamma \log N$ with $\gamma = -0.5$. Additionally, $\theta_k^* \in \mathbb{R}^{p+1}$ is generated from a $(p+1)$–dimension standard normal distribution for $1 \leq k \leq K$. To gain some intuitive understanding, we report in Table 1 the estimated values for $E(N_k)$ and $E(N_k)/N$. We find that $E(N_k)/N$ steadily decreases towards 0 as $N \to \infty$. Meanwhile, we find $E(N_k) \to \infty$ as $N \to \infty$.

*Table 1.* Rare class phenomenon with different $p$ and $K$. Recall $N$ is the total sample size. Define $E(N_k)$ as the expected average sample size of $K$ rare classes, and $E(N_k)/N$ as the rare class percentage.

| $N$ | $p = 10, K = 10$ | | $p = 50, K = 20$ | | $p = 500, K = 50$ | |
|---|---|---|---|---|---|---|
| | $E(N_k)$ | $E(N_k)/N(\%)$ | $E(N_k)$ | $E(N_k)/N(\%)$ | $E(N_k)$ | $E(N_k)/N(\%)$ |
| $10^5$ | 507 | 0.507 | 472 | 0.472 | 412 | 0.412 |
| $2 \times 10^5$ | 730 | 0.365 | 688 | 0.344 | 622 | 0.311 |
| $5 \times 10^5$ | 1,172 | 0.234 | 1,120 | 0.224 | 1,045 | 0.209 |

To evaluate the performance of the GMLE and PMLE, different numbers of rare classes (i.e., $K$) and different feature dimensions (i.e., $p$) are studied in three cases. For GMLE, the NR method and the gradient descent (GD) method are studied. We start with a relatively simple case with a small number of rare classes $K = 10$ and a low feature dimension $p = 10$. Let $\widehat{\Theta}^{(m)} = (\widehat{\theta}_{k,j}^{(m)} : 1 \leq j \leq p + 1, 1 \leq k \leq K)^{\top}$ be one particular estimator obtained in the $m$–th replication (e.g., $\widehat{\Theta}^{\text{pair}}$). To evaluate the estimation accuracy, we calculate the Root Mean Square Error (RMSE) as $\text{RMSE}^{(m)} = \{K^{-1}(p+1)^{-1} \sum_{k=1}^{K} \sum_{j=1}^{p+1} (\widehat{\theta}_{k,j}^{(m)} - \theta_{k,j})^2\}^{1/2}$ for the $m$–th replication. This leads to a total of $M = 100$ RMSE values. Those RMSE values are then log-transformed and box-plotted in Figure 3(a) and (b). To evaluate the computation efficiency, the average CPU time values (multiplied by 10 and log-transformed) for each method are reported.

By Figure 3(a), we find that log(RMSE) values of the PMLE

are almost the same as those of the GMLE. Moreover, a larger sample size $N$ makes these three estimators more accurate with smaller log(RMSE) values. The nearly identical results of both the NR and GD methods for GMLE suggest that both algorithms can be used to compute the GMLE. The nearly identical results between PMLE and GMLE (both NR and GD) suggest that the PMLE should be asymptotically as efficient as the GMLE. All these findings are consistent with our theoretical claims. By Figure 3(b), the average CPU time values required for these three methods increase as the sample size $N$ increases. Compared to the NR algorithm, the GD algorithm requires more time for GMLE computation. Among all these three methods, we find that the PMLE costs the least amount of computation time.

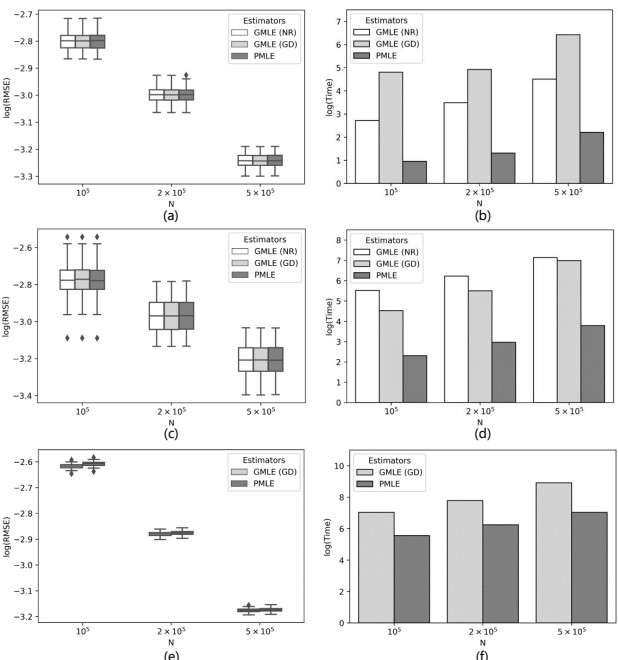

*Figure 3.* The left panel shows the boxplots of RMSE values in log-scale for the three estimation methods. The right panel presents the bar charts of the average CPU time values in log-scale for each method. The left white box shows the results for the GMLE computed by the NR algorithm. The middle light box illustrates the results for the GMLE computed by the GD algorithm. The right dark box presents the results for the PMLE by the NR algorithm. Each box is summarized based on $M = 100$ replications.

### 3.2. Larger $p$ and $K$

We next consider cases with a larger number of rare classes (i.e., $K = 20$) and a higher feature dimension (i.e., $p = 50$). In this case, the global Hessian matrix dimension becomes $(p+1)K \times (p+1)K = 1,020 \times 1,020$ (including the intercept term). Thus, it is a more challenging case for the NR algorithm, even though it remains implementable. The

resulting log(RMSE) values and average CPU time values in log-scale are computed as before and are plotted in Figure 3(c) and (d). By Figure 3(c), we find that log(RMSE) values of these three methods are almost the same. A larger $N$ makes these three estimators more accurate with steadily decreasing log(RMSE) values. These findings are consistent with our theoretical finding. As shown in Figure 3(d), the computational CPU time values of these three methods escalates with the growth of the sample size $N$. However, the story changes for the NR algorithm and the GD algorithm for the GMLE. With the global Hessian matrix dimension expanding to $1,020 \times 1,020$, the NR algorithm becomes more expensive in the CPU time cost as compared with the GD algorithm. Moreover, it is observed that the computational efficiency of the PMLE remains the best.

We next study an even more challenging case with a much larger number of rare classes ($K = 50$) and a much higher feature dimension ($p = 500$). In this case, the global Hessian matrix dimension becomes $(p + 1)K \times (p + 1)K = 25,050 \times 25,050$. The standard NR algorithm can hardly be used to compute the GMLE in this case. Therefore, a standard GD method is used to compute the GMLE. However, the PMLE can still be easily computed on a parallel or distributed system. The resulting log(RMSE) values and average CPU time values in log-scale are computed as before and are then plotted in Figure 3(e) and (f). By Figure 3(e), we find that log(RMSE) values of PMLE are slightly larger than GMLE. The difference of log(RMSE) values between these two methods vanishes as $N$ becomes larger. These findings are consistent with our theoretical finding that both the GMLE and PMLE are consistent estimators with the same asymptotic efficiency. As shown in Figure 3(f), the PMLE remains computationally more efficient than the GMLE computed by the GD algorithm in the term of the CPU time cost.

*Table 2.* Simulation results with $p = 50$ and $K = 20$. Define $E(N_0)$ as the expected sample size of the major class used for PMLE, $E(N_0 b_i)$ as the expected sample size of the major class used for SPMLE, and $E(N_k)$ as the expected average sample size of $K$ rare classes.

| $N$ | $E(N_0)$ | $E(N_0 b_i)$ | $E(N_k)$ | RMSE$_{\text{PMLE}}$ | RMSE$_{\text{SPMLE}}$ | T$_{\text{PMLE}}$ | T$_{\text{SPMLE}}$ |
|---|---|---|---|---|---|---|---|
| $10^5$ | 90,558 | 28,603 | 472 | 0.061 | 0.062 | 0.875 | 0.281 |
| $2 \times 10^5$ | 186,241 | 54,930 | 688 | 0.050 | 0.050 | 1.698 | 0.489 |
| $5 \times 10^5$ | 477,607 | 128,541 | 1,120 | 0.039 | 0.039 | 3.959 | 0.895 |

### 3.3. PMLE with Subsampling

As mentioned before, the SPMLE method adopts a much reduced sample size for the major class, so that the computation cost can be further reduced. We then study the performance of the SPMLE. For illustration, we fix $p = 50$ and $K = 20$. The total sample sizes are set to be $N = $

$1 \times 10^5, 2 \times 10^5$ and $5 \times 10^5$. For a fixed $N$, we set different sampling probabilities as $\pi_N = N^{-0.1}$. This satisfies the condition in Theorem 2.3 that $\pi_N / e^{\alpha_N} = N^{0.4} \to \infty$ as $N \to \infty$. We report $E(N_0 b_i)$ values as the estimated sample sizes of the major class used for SPMLE, which are much smaller than the estimated sample sizes of the major class used for PMLE $E(N_0)$. The average RMSE values are computed as before and reported as RMSE$_{\text{SPMLE}}$ in Table 2. The average CPU time values are computed as before and reported as T$_{\text{SPMLE}}$ in Table 2. For comparison, the average RMSE values and average CPU time values of PMLE are also provided as RMSE$_{\text{PMLE}}$ and T$_{\text{PMLE}}$. By Table 2, we find that the average RMSE values of SPMLE decrease as $N$ becomes larger. Meanwhile, we find that the average RMSE values of the SPMLE are slightly larger than those of the PMLE but with much reduced computation cost. These findings are consistent with our theoretical finding that the SPMLE is a consistent estimator with the same asymptotic efficiency as the PMLE.

## 4. TikTok Screenshots Dataset

### 4.1. TikTok Screenshots Dataset

We present in this section an interesting real data example. The dataset contains a total of 2,559 screenshots of size $720 \times 1,280$ randomly taken from TikTok live streams sponsored by different Audi dealers in China. For convenience, we refer to this dataset as the TikTok Screenshots (TTS) dataset. For a reliable evaluation, we randomly split the entire TTS dataset into two parts. The first part contains a total of 2,047 high-resolution images (about 80% of the whole data) for training, while the remaining 512 high-resolution images (about 20% of the whole data) are reserved for testing. For an intuitive understanding, randomly selected screenshot images are given in Figure 4(a). As can be seen, the screenshot images usually take an Audi car in the center place with detailed car model information displayed in the car plate; see the red box in Figure 4(b). A total of eight car models (i.e., classes) are included in the TTS dataset. They are, respectively, A3 Sportback, A4L, A5 Sportback, A6L, A7 Sportback, Q5L Sportback, Q7, and RS5. Then the objective of this study is to analyze the graphical information provided by the car plate so that the car model can be automatically and accurately recognized.

To prepare the data for subsequent analysis, we annotate each screenshot image manually with a tight bounding box for each car plate; see the red box shown in Figure 4(b). Next, a large number of $100 \times 40$ boxes are randomly generated within the image; see the blue boxes in Figure 4(c). Then the sub-images bordered by these blue boxes are extracted and saved on a hard drive. We treat each sub-image as one sample. For each sample (i.e., sub-image), we need to calculate its intersection-over-union (IoU) score

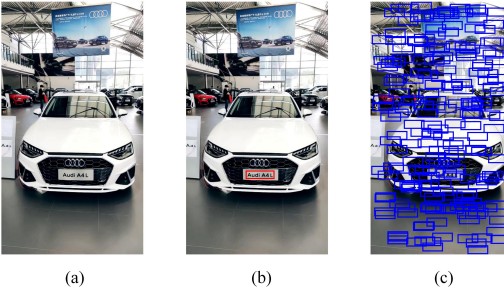

(a)       (b)       (c)

*Figure 4.* Left panel presents an example of the original image with size $720 \times 1,280$; middle panel presents the annotated image with the car plate tightly bounded by a red bounding box; right panel presents a total of 200 randomly generated blue boxes with size $100 \times 40$.

with the bounding box (Everingham et al., 2010). Here the IoU is defined as the area of $\mathbb{R} \cap \mathbb{B}$ over the area of $\mathbb{R} \cup \mathbb{B}$, where $\mathbb{R}$ refers to the sub-image contained in the red box and $\mathbb{B}$ refers to the sub-image contained in the blue box. Following Redmon et al. (2016) and Liu et al. (2016), we label the extracted sub-image as 0 if the corresponding IoU $< 0.2$. In contrast, we label it as a class $k$ with $1 \leq k \leq K$ according to the model type indicated by the car plate if the IoU $> 0.8$. The sub-images with $0.2 \leq \text{IoU} \leq 0.8$ are discarded. See Figure 5 for an intuitive understanding of the extracted sub-images and their corresponding class labels. This leads to a total of $N = 56,016$ sub-images belonging to nine classes, where the first class (i.e., $k = 0$) represents the background class; the remainder are car model classes. The major class (i.e., $k = 0$) accounts for more than 95% of the total sample. The percentages of other classes are bar-plotted in Figure 6 of Appendix A.4. As shown, this dataset aligns well with our theoretical framework.

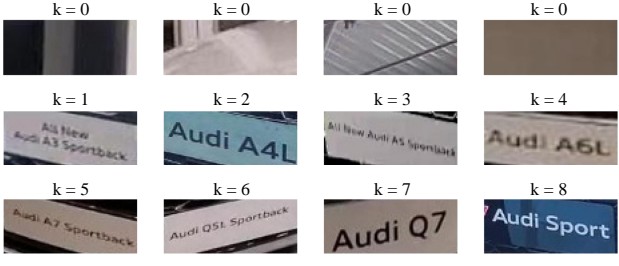

*Figure 5.* Examples of sub-images extracted from the TTS dataset. The class label is given above each sub-image. Here $k = 0$ (background) is the major class. All other classes represent different car models. Specifically, $k = 1$ (A3 Sportback), $k = 2$ (A4L), $k = 3$ (A5 Sportback), $k = 4$ (A6L), $k = 5$ (A7 Sportback), $k = 6$ (Q5L Sportback), $k = 7$ (Q7) and $k = 8$ (RS5).

We next consider how to represent each extracted sub-image by a feature vector. In this regard, we follow the transfer learning idea of Tan et al. (2018) and Zhuang et al. (2020). Specifically, we directly transfer a classical deep

learning model (i.e., the Visual Geometry Group's VGG16 (Simonyan & Zisserman, 2014)), which is a classic convolutional neural network with 20 layers and 14,714,688 parameters. Given the limited sample size, we cannot estimate those parameters accurately by using our dataset. Instead, we directly adopt the pre-trained estimates from the ImageNet dataset, which has 1,341,167 images (Deng et al., 2009). With the pre-trained parameter estimates, we could use the VGG16 model to convert each image into a feature vector of 512 dimensions. We finally arrive at a complete dataset with a sample size of $N = 56,016$, $p = 512$ feature dimension and $(K + 1) = 9$ classes.

### 4.2. Performance Evaluation

We first consider how to compute the GMLE. As mentioned before, the current dataset has high dimensional features with $p = 512$ and a total of $(K + 1) = 9$ classes. Consequently, the standard NR algorithm can hardly be used to compute the GMLE. To solve this problem, we consider the GD method (Dekel et al., 2012). Various learning rates are considered and the best learning rate is used. We next consider how to compute the PMLE. The PMLE is computationally convenient for two reasons. First, for each class pair, the Hessian matrix of the pairwise loss function is only of dimensions $513 \times 513$, which can be easily inverted. Subsequently, the NR algorithm can be readily implemented without any tuning parameter (e.g., the learning rate) selection issues. The algorithm converges extremely fast with a quadratic convergence rate, which is much larger than the linear convergence rate of a standard GD method (Curry, 1944). Second, by the pairwise likelihood, the original multi-class learning problem is divided into multiple two-class learning problems. This problem can be trained in a fully parallel or distributed manner. That makes distributed estimation feasible and then leads to a significant reduction in time cost. Lastly, the SPMLE method is also implemented with the sampling probability given as $\pi_N = 0.25$. By doing so, a much reduced sample size for the major class can be used and the computation cost can be further reduced.

Additionally, we have included the following competitive methods for comparison on the TTS dataset: the focal loss (FL) of Lin et al. (2017), the class-balanced loss (CBL) of Cui et al. (2019), the cost sensitive loss (CSL) and random downsampling (RDS) of Fernández et al. (2018). Specifically, FL introduces a modulating term to the cross-entropy loss, focusing on hard samples and downweighting easy negatives. CBL is proposed to reweight the loss inversely proportional to the effective sample count per class. CSL addresses the imbalanced distribution by adjusting misclassification costs. RDS is a technique that randomly sample the instances in the major class to balance the class distribution. All methods are optimized according to the suggestions of the original papers. Code is available at

*Table 3.* Prediction results for the TTS dataset.

|      | GMLE  | PMLE  | SPMLE | FL    | CBL   | CSL   | RDS   |
|------|-------|-------|-------|-------|-------|-------|-------|
| ACC  | 0.836 | 0.835 | 0.824 | 0.794 | 0.789 | 0.747 | 0.763 |
| AUC  | 0.997 | 0.999 | 0.999 | 0.998 | 0.998 | 0.991 | 0.996 |

Once an estimator (e.g., the GMLE) is computed, it is then applied to the testing dataset for the prediction evaluation. In this case, let $\widehat{\Theta} = \{\widehat{\theta}_k : 1 \le k \le K\}$ be the estimator computed from the training data and $\{(X_i^*, Y_i^*) : 1 \le i \le N^*\}$ be the testing data. Then we can compute the prediction probability $\widehat{w}_{i0} = 1/\{1 + \sum_{k=1}^K \exp(X_i^{*\top}\widehat{\theta}_k)\}$ and $\widehat{w}_{ik} = \exp(X_i^{*\top}\widehat{\theta}_k)/\{1 + \sum_{k=1}^K \exp(X_i^{*\top}\widehat{\theta}_k)\}$ for $1 \le k \le K$. We next follow the idea of Qiao & Liu (2009) and predict the response by $\widehat{Y}_i = \mathrm{argmax}_{0 \le k \le K} \widehat{w}_{ik}/\widetilde{\pi}_k$, where $\widetilde{\pi}_k$ is the $k$–th class percentage of the training dataset. Then the prediction accuracy can be computed as $\mathrm{ACC}_k = N_k^{*-1} \sum_{i=1}^{N^*} I(Y_i^* = k) \times I(\widehat{Y}_i = k)$ with the $k$–th sample size $N_k^* = \sum_{i=1}^{N^*} I(Y_i^* = k)$ for $0 \le k \le K$. The overall accuracy is then calculated as $\mathrm{ACC} = (K+1)^{-1} \sum_{k=0}^K \mathrm{ACC}_k$. Next, for the $k$–th class, we pair each binary response $\widetilde{Y}_i = I(Y_i^* = k)$ with its estimated class probability $\widehat{w}_{ik}$ and the $\mathrm{AUC}_k$ value (Ling et al., 2003) can be computed. The overall AUC value is then calculated as $\mathrm{AUC} = (K+1)^{-1} \sum_{k=0}^K \mathrm{AUC}_k$. Both the values for ACC and AUC are reported in Table 3.

By Table 3, we find that our methods outperform all competitors. Among our methods, we observe that the overall ACC value of the GMLE method is 0.836, which is slightly higher than 0.835 of the PMLE method and 0.824 of the SPMLE method. In terms of AUC values, all the three methods are extremely similar with very tiny difference, which might be due to random error. The key difference is the computational time. More specifically, the CPU time cost of GMLE is about 145.85 seconds. In contrast, those of PMLE and SPMLE are given as 42.12 seconds and 12.12 seconds, respectively. The outstanding computational efficiency of PMLE and SPMLE is mainly because they can be computed by the NR algorithm in a parallel way. The computation cost of the SPMLE method is lower due to a much reduced sample size for the major class.

## 5. Concluding Remarks

We study the problem of multi-class logistic regression with one major class and multiple rare classes, which is motivated by a real application about TTS data analysis. To motivate our method, we make a natural extension from the two-class logistic model in Wang (2020) and construct a multi-class logistic regression model with one major class and multiple rare classes. Next, we conduct a rigorous asymptotic study of the standard MLE. Based on the special structure of the MLE's asymptotic covariance matrix, we propose the PMLE. To further reduce the computational cost, we propose the SPMLE by down-sampling the major class. Theoretically, the PMLE and SPMLE are statistically efficient asymptotically. Both of them are suitable for distributed computation with lower computational cost. Moreover, the SPMLE is easier to compute with much less computation cost compared with the other two estimators. Extensive simulation studies and TikTok live stream data analysis are presented to demonstrate their finite sample performance. To conclude this work, we discuss some interesting directions for future research. First, we focus on the logistic regression model in this paper. It is interesting to investigate more complicated and general models for the multi-class data with one major class and multiple rare classes. Second, the performance of our method in a situation that the sample sizes of the different rare classes are of different orders remains unknown and requires exploration. Therefore, how to extend these methods to more flexible settings is an interesting direction for future work.

## Acknowledgements

We thank the anonymous reviewers for helpful comments on earlier versions of this paper. Danyang Huang's research is partially supported by the National Natural Science Foundation of China (72471230), fund for building world-class universities (disciplines) of the Renmin University of China, the MOE Project of Key Research Institute of Humanities and Social Sciences (grant 22JJD110001) and Public Computing Cloud, Renmin University of China. Hansheng Wang's research is partially supported by National Natural Science Foundation of China (72495123, 12271012).

## Impact Statement

This paper addresses the problem of multi-class logistic regression with a major class and multiple rare classes. We propose the PMLE and SPMLE methods. Unlike the computational challenges of traditional approaches, they reduce complexity by decomposing the problem into multiple pairwise two-class problems, while ensuring asymptotic efficiency and enabling parallel computation. This work provides an efficient solution for handling imbalanced multi-class data and contributes to related fields in machine learning.

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

# A. Supplementary Material

### A.1. Proof of Theorem 2.1

To study the asymptotic property of MLE, we first derive the log-likelihood function $\mathcal{L}(\Theta^*) = \sum_{i=1}^{N} \sum_{k=0}^{K} a_{ik} \log w_{ik}^*$ based on (2), where $\Theta^* = (\theta_1^{*\top}, \cdots, \theta_K^{*\top})^\top \in \mathbb{R}^q$ is the stacked coefficient vector with dimension $q = (p+1)K$ and $\theta_k = (\beta_{k0}, \beta_k^\top)^\top \in \mathbb{R}^{p+1}$ be the associated regression coefficient parameter. Here $a_{ik} = I(Y_i = k) \in \{0, 1\}$ is the binary vector and $w_{ik}^* = P(Y_i = k | Z_i)$ is the rare class response probability vector. Next, we can estimate the maximum likelihood estimator $\widetilde{\Theta} = \operatorname{argmax}_{\Theta^*} \mathcal{L}(\Theta^*)$. Specifically, to show that $\widetilde{\Theta}$ is $\sqrt{Ne^{\alpha_N}}$–consistent, it suffices to verify there exists some constant $C > 0$ such that

$$\sup_{\xi \in \mathbb{R}^u, \|\xi\| = C} \mathcal{L}(\Theta^* + \xi/\sqrt{Ne^{\alpha_N}}) < \mathcal{L}(\Theta^*) \tag{5}$$

with probability tending to 1 as $N$ goes to infinity (Fan & Li, 2001). By Taylor's expansion, we have

$$\mathcal{L}(\Theta^* + \xi/\sqrt{Ne^{\alpha_N}}) - \mathcal{L}(\Theta^*) = \left\{ \xi^\top \dot{\mathcal{L}}(\Theta^*)/\sqrt{Ne^{\alpha_N}} + \xi^\top \ddot{\mathcal{L}}(\Theta^*)\xi/(2Ne^{\alpha_N}) \right\} \left\{ 1 + o_p(1) \right\}.$$

We can demonstrate that $\dot{\mathcal{L}}(\Theta^*)/\sqrt{Ne^{\alpha_N}} = O_p(1)$ in Step 1, and $\ddot{\mathcal{L}}(\Theta^*)/(Ne^{\alpha_N})$ converges to a negative definite constant matrix in probability in Step 2. Then we could verify (5) holds for a sufficiently large $C$. Due to the strict convexity of $\mathcal{L}(\Theta^*)$, we obtain that $\sup_{\|\xi\| > C} \mathcal{L}(\Theta^* + \xi/\sqrt{Ne^{\alpha_N}}) < \mathcal{L}(\Theta^*)$. As $\mathcal{L}(\Theta^*)$ is maximized at $\widetilde{\Theta}$, we know $\widetilde{\Theta}$ lies in the ball $\{\Theta^* + \xi/\sqrt{Ne^{\alpha_N}} : \|\xi\| \le C\}$. In other words, we have $\widetilde{\Theta} = O_p(1/\sqrt{Ne^{\alpha_N}})$.

Given $\widetilde{\Theta}$ is $\sqrt{Ne^{\alpha_N}}$–consistent, we can apply the Taylor expansion and have

$$\sqrt{Ne^{\alpha_N}}(\widetilde{\Theta} - \Theta^*) = \left\{ \ddot{\mathcal{L}}^{-1}(\widetilde{\Theta}^*)/(Ne^{\alpha_N}) \right\}^{-1} \left\{ \dot{\mathcal{L}}(\Theta^*)/\sqrt{Ne^{\alpha_N}} \right\},$$

where $\widetilde{\Theta}^*$ lies between $\widetilde{\Theta}$ and $\Theta^*$. In the following two steps, we would further prove when $N \to \infty$, we have

$$\left(Ne^{\alpha_N}\right)^{-1/2} \dot{\mathcal{L}}(\Theta^*) \to_d N(0, \Sigma^*), \tag{6}$$

$$\left(Ne^{\alpha_N}\right)^{-1} \ddot{\mathcal{L}}(\Theta^*) \to_p -\Sigma^*, \tag{7}$$

where $\Sigma^* = \operatorname{diag}\{\Sigma_k^* : 1 \le k \le K\}$ and $\Sigma_k^* = E\{\exp(Z_i^\top \theta_k^*) Z_i Z_i^\top\}$. This completes the proof of Theorem 2.1.

STEP 1. To verify (6), we compute the mean and covariance of $(Ne^{\alpha_N})^{-1/2} \dot{\mathcal{L}}(\Theta^*)$ respectively. For simplicity, we denote $\dot{\mathcal{L}}_k(\Theta^*) = \partial \mathcal{L}(\Theta^*)/\partial \theta_k^*$. Then, we can rewrite $\dot{\mathcal{L}}(\Theta^*) = (\dot{\mathcal{L}}_1(\Theta^*)^\top, \cdots, \dot{\mathcal{L}}_K(\Theta^*)^\top)^\top \in \mathbb{R}^q$. Since we have $\dot{\mathcal{L}}_k(\Theta^*) = \sum_{i=1}^{N} (a_{ik} - w_{ik}^*) Z_i$, it suffices to compute the mean and covariance of $(Ne^{\alpha_N})^{-1/2} \dot{\mathcal{L}}_k(\Theta^*)$ respectively.

First, we compute $E\{(Ne^{\alpha_N})^{-1/2} \dot{\mathcal{L}}_k(\Theta^*)\}$. It can be verified that $E\{\dot{\mathcal{L}}_k(\Theta^*)\} = NE[\{E(a_{ik}|Z_i) - w_{ik}^*\} Z_i] = 0$. Hence, we prove $E\{\dot{\mathcal{L}}(\Theta^*)\} = 0$. Next, we calculate $\operatorname{cov}\{\dot{\mathcal{L}}(\Theta^*)\}$. Then, it suffices to work on the diagonal matrix $\operatorname{cov}\{(Ne^{\alpha_N})^{-1/2} \dot{\mathcal{L}}_k(\Theta^*)\}$ and the off-diagonal matrix $\operatorname{cov}\{(Ne^{\alpha_N})^{-1/2} \dot{\mathcal{L}}_{k_1}(\Theta^*), (Ne^{\alpha_N})^{-1/2} \dot{\mathcal{L}}_{k_2}(\Theta^*)\}$, respectively. One can verify that

$$\operatorname{cov}\left\{ (Ne^{\alpha_N})^{-1/2} \dot{\mathcal{L}}_k(\Theta^*) \right\} = e^{-\alpha_N} \operatorname{cov}\left\{ (a_{ik} - w_{ik}^*) Z_i \right\} = E\left\{ \frac{\exp(Z_i^\top \theta_k^*)}{1 + \sum_{k=1}^{K} \exp(Z_i^\top \theta_k^* + \alpha_N)} Z_i Z_i^\top \right\}$$

$$= -e^{\alpha_N} E\left\{ \frac{\exp(2Z_i^\top \theta_k^*)}{\{1 + \sum_{k=1}^{K} \exp(Z_i^\top \theta_k^* + \alpha_N)\}^2} Z_i Z_i^\top \right\} = E\left\{ \exp(Z_i^\top \theta_k^*) Z_i Z_i^\top \right\} \{1 + o(1)\}. \tag{8}$$

To obtain the first equality of (8), we use the fact that $\operatorname{cov}\{(a_{ik} - w_{ik}^*) Z_i\} = E[E\{(a_{ik} - w_{ik}^*)^2 | Z_i\} Z_i Z_i^\top] = E(w_{ik}^* Z_i Z_i^\top) - E(w_{ik}^{*2} Z_i Z_i^\top)$. Consequently, we find that the second term $E(w_{ik}^{*2} Z_i Z_i^\top)$ is the higher order term. Thus, we focus on the first term $E(w_{ik}^* Z_i Z_i^\top)$ and prove the second equality of (8). To this end, we need to verify the following three conditions. First, it is obvious that $\exp(Z_i^\top \theta_k^*)\{1 + \sum_{k=1}^{K} \exp(Z_i^\top \theta_k^* + \alpha_N)\}^{-1} \|Z_i\|^2 \le \exp(Z_i^\top \theta_k^*) \|Z_i\|^2$. Second, it is noteworthy that $\exp(Z_i^\top \theta_k^*)\{1 + \sum_{k=1}^{K} \exp(Z_i^\top \theta_k^* + \alpha_N)\}^{-1} Z_i Z_i^\top$ converges to $\exp(Z_i^\top \theta_k^*) Z_i Z_i^\top$ almost

surely as $N \to \infty$. Lastly, we know that $E\big(e^{Z_i^\top \theta_k^*}\|Z_i\|^2\big) < \infty$ by the condition that $E\big(e^{t\|Z_i\|}\big) < \infty$ for any $t > 0$ (Wang, 2020). Combining the above results, the second equality of (8) can be verified by the dominated convergence theorem. Then, by the similar technique, we calculate

$$
\begin{aligned}
& \text{cov}\Big\{(Ne^{\alpha_N})^{-1/2}\dot{\mathcal{L}}_{k_1}(\Theta^*), (Ne^{\alpha_N})^{-1/2}\dot{\mathcal{L}}_{k_2}(\Theta^*)\Big\} \\
=\ & e^{-\alpha_N} E\Big[E\big\{(a_{ik_1} - w_{ik_1}^*)(a_{ik_2} - w_{ik_2}^*) \mid Z_i\big\}Z_i Z_i^\top\Big] \\
=\ & -e^{\alpha_N} E\left[\frac{\exp\big(Z_i^\top \theta_{k_1}^* + Z_i^\top \theta_{k_2}^*\big)}{\big\{1 + \sum_{k=1}^K \exp\big(Z_i^\top \theta_k^* + \alpha_N\big)\big\}^2} Z_i Z_i^\top\right] \\
=\ & -e^{\alpha_N} E\big\{\exp\big(Z_i^\top \theta_{k_1}^* + Z_i^\top \theta_{k_2}^*\big)Z_i Z_i^\top\big\}\big\{1 + o(1)\big\} = O(e^{\alpha_N}),
\end{aligned}
$$

where the first equality is due to $E(a_{ik} - w_{ik}^*|Z_i) = 0$, and the second equality is because $E\big\{(a_{ik_1} - w_{ik_1}^*)(a_{ik_2} - w_{ik_2}^*)|Z_i\big\} = E\big(a_{ik_1}w_{ik_2}^* - w_{ik_1}^*a_{ik_2} + w_{ik_1}^*w_{ik_2}^*|Z_i\big) = -w_{ik_1}^*w_{ik_2}^*$. It refers that the off-diagonal matrix of $\text{cov}\big\{\sqrt{Ne^{\alpha_N}}\dot{\mathcal{L}}(\Theta^*)\big\}$ is of the order $O(e^{\alpha_N})$ and converges to 0 when $N \to \infty$.

We then verify the Lindeberg–Feller condition. Define $\eta_i = \big((a_{i1} - w_{i1}^*)Z_i^\top, \cdots, (a_{iK} - w_{iK}^*)Z_i^\top\big)^\top \in \mathbb{R}^q$. We then have

$$
\begin{aligned}
& \sum_{i=1}^N E\Big\{\|\eta_i\|^2 I\big(\|\eta_i\| > \sqrt{Ne^{\alpha_N}}\epsilon\big)\Big\} \\
=\ & NE\bigg[\sum_{k=1}^K w_{ik}^*\Big\{\big\|(1 - w_{ik}^*)Z_i\big\|^2 + \sum_{k' \neq k}\big\|w_{ik'}^*Z_i\big\|^2\Big\}I\Big(\big\|(1 - w_{ik}^*)Z_i\big\|^2 + \sum_{k' \neq k}\big\|w_{ik'}^*Z_i\big\|^2 > Ne^{\alpha_N}\epsilon^2\Big)\bigg] \\
& + NE\Big\{w_{i0}^*\sum_{k=1}^K\big\|w_{ik}^*Z_i\big\|^2 I\big(\sum_{k=1}^K\big\|w_{ik}^*Z_i\big\|^2 > Ne^{\alpha_N}\epsilon^2\big)\Big\}.
\end{aligned}
$$

We first compute that

$$
\begin{aligned}
& NE\bigg[\sum_{k=1}^K w_{ik}^*\Big\{\big\|(1 - w_{ik}^*)Z_i\big\|^2 + \sum_{k' \neq k}\big\|w_{ik'}^*Z_i\big\|^2\Big\}I\Big(\big\|(1 - w_{ik}^*)Z_i\big\|^2 + \sum_{k' \neq k}\big\|w_{ik'}^*Z_i\big\|^2 > Ne^{\alpha_N}\epsilon^2\Big)\bigg] \\
\leq\ & NE\Big\{\sum_{k=1}^K w_{ik}^*\big(\sum_{k=1}^K w_{ik}^{*2} + 1\big)\|Z_i\|^2 I\big(\sum_{k=1}^K\big\|w_{ik}^*Z_i\big\|^2 > Ne^{\alpha_N}\epsilon^2\big)\Big\} \\
\leq\ & Ne^{\alpha_N}E\big\{\sum_{k=1}^K e^{Z_i^\top \theta_k^*}\big(\sum_{k=1}^K w_{ik}^{*2} + 1\big)\|Z_i\|^2 I\big(\sum_{k=1}^K\big\|w_{ik}^*Z_i\big\|^2 > Ne^{\alpha_N}\epsilon^2\big)\big\} = o\big(Ne^{\alpha_N}\big).
\end{aligned}
$$

Moreover, it is can be verified that

$$
\begin{aligned}
& NE\Big\{w_{i0}^*\sum_{k=1}^K\big\|w_{ik}^*Z_i\big\|^2 I\big(\sum_{k=1}^K\big\|w_{ik}^*Z_i\big\|^2 > Ne^{\alpha_N}\epsilon^2\big)\Big\} \\
\leq\ & NE\Big\{\sum_{k=1}^K\big\|w_{ik}^*Z_i\big\|^2 I\big(\sum_{k=1}^K\big\|w_{ik}^*Z_i\big\|^2 > Ne^{\alpha_N}\epsilon^2\big)\Big\} = o\big(Ne^{\alpha_N}\big).
\end{aligned}
$$

Thus, for any $\epsilon > 0$, we have $\sum_{i=1}^N E\big\{\|\eta_i\|^2 I\big(\|\eta_i\| > \sqrt{Ne^{\alpha_N}}\epsilon\big)\big\} = o\big(Ne^{\alpha_N}\big)$. Combining the above results, we finish the proof of (6) by applying the Lindeberg–Feller central limit theorem (Van der Vaart, 2000).

STEP 2. To verify (7), we compute the mean and covariance of $(Ne^{\alpha_N})^{-1}\ddot{\mathcal{L}}(\Theta^*)$ respectively. For simplicity, we denote the $k$–th diagonal block matrix $\ddot{\mathcal{L}}_k(\Theta^*) = \partial^2\mathcal{L}(\Theta^*)/\partial\theta_k^*\partial\theta_k^{*\top} \in \mathbb{R}^{(p+1)\times(p+1)}$ for $1 \leq k \leq K$, and denote the $(k_1, k_2)$–th off-diagonal block matrix $\ddot{\mathcal{L}}_{k_1,k_2}(\Theta^*) = \partial^2\mathcal{L}(\Theta^*)/\partial\theta_{k_1}^*\partial\theta_{k_2}^{*\top} \in \mathbb{R}^{(p+1)\times(p+1)}$ for $1 \leq k_1 \neq k_2 \leq K$.

First, we compute the mean and covariance of $(Ne^{\alpha_N})^{-1}\ddot{\mathcal{L}}_k(\Theta^*)$ respectively for $1 \le k \le K$. We can derive $\ddot{\mathcal{L}}_k(\Theta^*) = -\sum_{i=1}^{N} w_{ik}^*(1 - w_{ik}^*)Z_i Z_i^\top$. Then, we have

$$E\left\{(Ne^{\alpha_N})^{-1}\ddot{\mathcal{L}}_k(\Theta^*)\right\} = -e^{-\alpha_N} E\left\{w_{ik}^*(1 - w_{ik}^*)Z_i Z_i^\top\right\}$$

$$= -E\left\{\frac{\exp(Z_i^\top \theta_k^*)}{1 + \sum_{k=1}^{K} \exp(Z_i^\top \theta_k^* + \alpha_N)}\left(1 - \frac{\exp(Z_i^\top \theta_k^* + \alpha_N)}{1 + \sum_{k=1}^{K} \exp(Z_i^\top \theta_k^* + \alpha_N)}\right)Z_i Z_i^\top\right\}$$

$$= -E\left\{\exp(Z_i^\top \theta_k^*)Z_i Z_i^\top\right\}\left\{1 + o(1)\right\},$$

where the last step holds by applying the dominated convergence theorem. Then, we compute the covariance of $(Ne^{\alpha_N})^{-1}\ddot{\mathcal{L}}(\Theta^*)$. Consider the $(j_1, j_2)$–th component of $\ddot{\mathcal{L}}_k(\Theta^*)$ with $1 \le j_1, j_2 \le p + 1$, we could verify

$$\text{var}\left\{(Ne^{\alpha_N})^{-1}\ddot{\mathcal{L}}_{k,j_1 j_2}(\Theta^*)\right\} = N^{-1}e^{-2\alpha_N}\text{var}\left\{w_{ik}^*(1 - w_{ik}^*)Z_{i,j_1}Z_{i,j_2}\right\}$$

$$= N^{-1}E\left\{\frac{\exp(2Z_i^\top \theta_k^*)}{\{1 + \sum_{k=1}^{K} \exp(Z_i^\top \theta_k^* + \alpha_N)\}^2}\left(1 - \frac{\exp(Z_i^\top \theta_k^* + \alpha_N)}{1 + \sum_{k=1}^{K} \exp(Z_i^\top \theta_k^* + \alpha_N)}\right)^2 Z_{i,j_1}^2 Z_{i,j_2}^2\right\}$$

$$-N^{-1}E^2\left\{\frac{\exp(Z_i^\top \theta_k^*)}{1 + \sum_{k=1}^{K} \exp(Z_i^\top \theta_k^* + \alpha_N)}\left(1 - \frac{\exp(Z_i^\top \theta_k^* + \alpha_N)}{1 + \sum_{k=1}^{K} \exp(Z_i^\top \theta_k^* + \alpha_N)}\right)Z_{i,j_1}Z_{i,j_2}\right\}$$

$$= N^{-1}\text{var}\left\{\exp(Z_i^\top \theta_k^*)Z_{i,j_1}Z_{i,j_2}\right\}\left\{1 + o(1)\right\},$$

where the last step is because of the dominated convergence theorem. Consequently, we know that $\text{var}\left\{(Ne^{\alpha_N})^{-1}\right.$ $\left.\ddot{\mathcal{L}}_{k,j_1 j_2}(\Theta^*)\right\} \to 0$ when $N \to \infty$. Together with $E\left\{(Ne^{\alpha_N})^{-1}\ddot{\mathcal{L}}_k(\Theta^*)\right\}$, we have $(Ne^{\alpha_N})^{-1}\ddot{\mathcal{L}}_k(\Theta^*) \to_p E\left\{\exp(Z_i^\top \theta_k^*)Z_i Z_i^\top\right\}$ as $N \to \infty$ for the $k$–th diagonal block matrix.

Next, we compute the mean and covariance of $(Ne^{\alpha_N})^{-1}\ddot{\mathcal{L}}_{k_1 k_2}(\Theta^*)$ respectively for $1 \le k_1 \ne k_2 \le K$. We can derive $\ddot{\mathcal{L}}_{k_1 k_2}(\Theta^*) = \sum_{i=1}^{N} w_{ik_1}^* w_{ik_2}^* Z_i Z_i^\top$. Then, we have

$$E\left\{(Ne^{\alpha_N})^{-1}\ddot{\mathcal{L}}_{k_1 k_2}(\Theta^*)\right\} = e^{-\alpha_N} E\left\{w_{ik_1}^* w_{ik_2}^* Z_i Z_i^\top\right\}$$

$$= e^{-\alpha_N} E\left[\frac{\exp(Z_i^\top \theta_{k_1}^* + Z_i^\top \theta_{k_2}^* + 2\alpha_N)}{\{1 + \sum_{k=1}^{K} \exp(Z_i^\top \theta_k^* + \alpha_N)\}^2}Z_i Z_i^\top\right]$$

$$= e^{\alpha_N} E\left\{\exp(Z_i^\top \theta_{k_1}^* + Z_i^\top \theta_{k_2}^*)Z_i Z_i^\top\right\}\left\{1 + o(1)\right\},$$

where the last step is due to the dominated convergence theorem. This refers that when $N \to \infty$, $E\left\{(Ne^{\alpha_N})^{-1}\ddot{\mathcal{L}}_{k_1 k_2}(\Theta^*)\right\} \to 0$. We then compute the covariance matrix of $(Ne^{\alpha_N})^{-1}\ddot{\mathcal{L}}_{k_1 k_2}(\Theta^*)$. Here we focus on the $(j_1, j_2)$–th component of off-diagonal matrix $(Ne^{\alpha_N})^{-1}\ddot{\mathcal{L}}_{k_1 k_2}(\Theta^*)$ with $1 \le j_1, j_2 \le p + 1$, we have

$$\text{var}\left\{(Ne^{\alpha_N})^{-1}\ddot{\mathcal{L}}_{k_1 k_2, j_1 j_2}(\Theta^*)\right\} = N^{-1}e^{-2\alpha_N}\text{var}\left(w_{ik_1}^* w_{ik_2}^* Z_{i,j_1}Z_{i,j_2}\right)$$

$$= N^{-1}e^{-2\alpha_N}E\left[\frac{\exp(2Z_i^\top \theta_{k_1}^* + 2Z_i^\top \theta_{k_2}^* + 4\alpha_N)}{\{1 + \sum_{k=1}^{K} \exp(Z_i^\top \theta_k^* + \alpha_N)\}^4}Z_{i,j_1}^2 Z_{i,j_2}^2\right]$$

$$-N^{-1}e^{-2\alpha_N}E^2\left[\frac{\exp(Z_i^\top \theta_{k_1}^* + Z_i^\top \theta_{k_2}^* + 2\alpha_N)}{\{1 + \sum_{k=1}^{K} \exp(Z_i^\top \theta_k^* + \alpha_N)\}^2}Z_{i,j_1}Z_{i,j_2}\right]$$

$$= N^{-1}e^{2\alpha_N}\text{var}\left\{\exp(Z_i^\top \theta_{k_1}^* + Z_i^\top \theta_{k_2}^*)Z_{i,j_1}Z_{i,j_2}\right\}\left\{1 + o(1)\right\},$$

where the last step is obtained by the dominated convergence theorem. Then we immediately know $\text{var}\left\{(Ne^{\alpha_N})^{-1}\right.$ $\left.\ddot{\mathcal{L}}_{k_1 k_2, j_1 j_2}(\Theta^*)\right\} \to 0$ when $N \to \infty$. Further, this implies the maximum likelihood estimator converges asymptotically to a matrix with block-diagonal structure. Combining the above results, we have $(Ne^{\alpha_N})^{-1}\ddot{\mathcal{L}}_{k_1 k_2}(\Theta^*) \to_p 0$ as $N \to \infty$ for the off-diagonal block matrix. Thus, we accomplish the proof of (7).

## A.2. Proof of Theorem 2.2

In this appendix, we study the properties of the pairwise maximum likelihood estimator $\widehat{\theta}_k^{\text{pair}}$ for $1 \leq k \leq K$. Specifically, we need to investigate a set of major–and–rare class pair logistic regression models as follows

$$P\Big(Y_i = k \mid Z_i, Y_i \in \{0, k\}\Big) = \frac{\exp(Z_i^\top \theta_k^* + \alpha_N)}{1 + \exp(Z_i^\top \theta_k^* + \alpha_N)}$$

for every rare class $1 \leq k \leq K$. Define $w_{i,0k} = P\big(Y_i = k \mid Z_i, Y_i \in \{0, k\}\big)$. Consequently, the conditional log-likelihood function for the two-class logistic regression problem can be written as $\mathcal{L}_{0k}(\theta_k^*) = \sum_{i=1}^N a_{ik} \log w_{i,0k} + a_{i0} \log(1 - w_{i,0k}) = \sum_{i=1}^N a_{ik}(Z_i^\top \theta_k^* + \alpha_N) - (a_{ik} + a_{i0}) \log\big\{1 + \exp(Z_i^\top \theta_k^* + \alpha_N)\big\}$. Then a two-class MLE can be obtained as $\widehat{\theta}_k^{\text{pair}} = \operatorname{argmax}_{\theta_k^*} \mathcal{L}_{0k}(\theta_k^*)$. Similarly with the proof of Theorem 2.1, to show that $\widehat{\theta}_k^{\text{pair}}$ is $\sqrt{Ne^{\alpha_N}}$–consistent and asymptotically normal, We would further prove in the following two steps

$$\big(Ne^{\alpha_N}\big)^{-1/2} \dot{\mathcal{L}}_{0k}\big(\theta_k^*\big) \to_d N(0, \Sigma_k^*), \tag{9}$$

$$\big(Ne^{\alpha_N}\big)^{-1} \ddot{\mathcal{L}}_{0k}\big(\theta_k^*\big) \to_p -\Sigma_k^*, \tag{10}$$

as $N \to \infty$ in the following two steps, where $\Sigma_k^* = E\big\{\exp(Z_i^\top \theta_k^*) Z_i Z_i^\top\big\}$. This completes the proof of Theorem 2.2.

PROOF OF (9). We compute the mean and covariance of $\dot{\mathcal{L}}_{0k}\big(\theta_k^*\big)$, respectively. Note that $\dot{\mathcal{L}}_{0k}\big(\theta_k^*\big) = \sum_{i=1}^N \big\{a_{ik} - (a_{ik} + a_{i0})w_{i,0k}\big\} Z_i$. Consequently, the mean of $\dot{\mathcal{L}}_{0k}\big(\theta_k^*\big)$ can be calculate as $E\{\dot{\mathcal{L}}_{0k}(\theta_k^*)\} = NE\big[\big\{a_{ik} - (a_{ik} + a_{i0})w_{i,0k}\big\} Z_i\big] = NE\big[\big\{w_{ik}^* - (w_{ik}^* + w_{i0}^*)w_{i,0k}\big\} Z_i\big] = 0$ because of the fact that $(w_{ik}^* + w_{i0}^*)w_{i,0k} = w_{ik}^*$. In the meanwhile, the covariance of $\dot{\mathcal{L}}_{0k}\big(\theta_k^*\big)$ can be calculated as

$$\operatorname{cov}\Big\{\big(Ne^{\alpha_N}\big)^{-1/2}\dot{\mathcal{L}}_{0k}\big(\theta_k^*\big)\Big\} = e^{-\alpha_N} E\Big[\big\{a_{ik} - (a_{ik} + a_{i0})w_{i,0k}\big\}^2 Z_i Z_i^\top\Big]$$

$$= e^{-\alpha_N} E\Big[\big\{a_{ik} + (a_{ik} + a_{i0})w_{i,0k}^2 - 2a_{ik}w_{i,0k}\big\} Z_i Z_i^\top\Big]$$

$$= e^{-\alpha_N} E\Big[\big\{w_{ik}^* + (w_{ik}^* + w_{i0}^*)w_{i,0k}^2 - 2w_{ik}^* w_{i,0k}\big\} Z_i Z_i^\top\Big]$$

$$= e^{-\alpha_N} E\big\{w_{ik}^*(1 - w_{i,0k}) Z_i Z_i^\top\big\} = \Sigma_k^*\{1 + o(1)\},$$

where the first equality is due to the fact that $E\big[\big\{a_{ik} - (a_{ik} + a_{i0})w_{i,0k}\big\} Z_i\big] = 0$, the third equality holds because $(w_{ik}^* + w_{i0}^*)w_{i,0k} = w_{ik}^*$ and the last equality is by the dominated convergence theorem.

We then verify the Lindeberg–Feller condition. For any $\epsilon > 0$,

$$\sum_{i=1}^N E\Big[\big\|\{a_{ik} - (a_{ik} + a_{i0})w_{i,0k}\} Z_i\big\|^2 I\big(\|\{a_{ik} - (a_{ik} + a_{i0})w_{i,0k}\} Z_i\| > \sqrt{Ne^{\alpha_N}}\epsilon\big)\Big]$$

$$= NE\Big\{w_{ik}^*(1 - w_{i,0k})^2 \|Z_i\|^2 I\big(\|(1 - w_{i,0k})Z_i\| > \sqrt{Ne^{\alpha_N}}\epsilon\big)\Big\}$$

$$+ NE\Big\{(1 - w_{ik}^*)w_{i,0k}^2 \|Z_i\|^2 I\big(\|w_{i,0k}Z_i\| > \sqrt{Ne^{\alpha_N}}\epsilon\big)\Big\}$$

$$\leq NE\Big\{w_{ik}^* \|Z_i\|^2 I\big(\|Z_i\| > \sqrt{Ne^{\alpha_N}}\epsilon\big)\Big\} + NE\Big\{w_{i,0k}^2 \|Z_i\|^2 I\big(\|w_{i,0k}Z_i\| > \sqrt{Ne^{\alpha_N}}\epsilon\big)\Big\}$$

$$\leq 2Ne^{\alpha_N} E\Big\{e^{\|Z_i^\top \theta_k^*\|} \|Z_i\|^2 I\big(\|Z_i\| > \sqrt{Ne^{\alpha_N}}\epsilon\big)\Big\} = o\big(Ne^{\alpha_N}\big).$$

Combining the above results, we finish the proof of (9) by applying the Lindeberg–Feller central limit theorem (Van der Vaart, 2000).

PROOF OF (9). We compute the mean and covariance of $\ddot{\mathcal{L}}_{0k}\big(\theta_k^*\big)$ respectively. Note that $\ddot{\mathcal{L}}_{0k}\big(\theta_k^*\big) = -\sum_{i=1}^N (a_{ik} + a_{i0})w_{i,0k}(1 - w_{i,0k}) Z_i Z_i^\top$. Hence, we obtain the mean of $\ddot{\mathcal{L}}_{0k}\big(\theta_k^*\big)$ as $E\{(Ne^{\alpha_N})^{-1}\ddot{\mathcal{L}}_{0k}\big(\theta_k^*\big)\} = -e^{-\alpha_N} E\{(a_{ik} + a_{i0})w_{i,0k}(1 - w_{i,0k}) Z_i Z_i^\top\} = -e^{-\alpha_N} E\{(w_{ik}^* + w_{i0}^*)w_{i,0k}(1 - w_{i,0k}) Z_i Z_i^\top\} = -e^{-\alpha_N} E\{w_{ik}^*(1 - w_{i,0k}) Z_i Z_i^\top\} =$

$-\Sigma_k^*\{1 + o(1)\}$. We then compute the variance of the $(j_1, j_2)$–th element of $(Ne^{\alpha_N})^{-1}\ddot{\mathcal{L}}_{0k}(\theta_k^*)$ and have

$$
\begin{aligned}
&\text{var}\Big\{\big(Ne^{\alpha_N}\big)^{-1}\ddot{\mathcal{L}}_{0k,j_1j_2}\big(\theta_k^*\big)\Big\} \\
=\ & N^{-1}e^{-2\alpha_N}E\Big\{\big(a_{ik} + a_{i0}\big)w_{i,0k}^2\big(1 - w_{i,0k}\big)^2 Z_{ij_1}^2 Z_{ij_2}^2\Big\} \\
& -N^{-1}e^{-2\alpha_N}E^2\Big\{\big(a_{ik} + a_{i0}\big)w_{i,0k}\big(1 - w_{i,0k}\big)Z_{ij_1}Z_{ij_2}\Big\} \\
=\ & N^{-1}e^{-2\alpha_N}\Big[E\Big\{w_{ik}^* w_{i,0k}\big(1 - w_{i,0k}\big)^2 Z_{ij_1}^2 Z_{ij_2}^2\Big\} - E^2\Big\{w_{ik}^*\big(1 - w_{i,0k}\big)Z_{ij_1}Z_{ij_2}\Big\}\Big] \\
=\ & N^{-1}\text{var}\Big\{\exp\big(Z_i^\top\theta_k^*\big)Z_{ij_1}Z_{ij_2}\Big\}\{1 + o(1)\} = O(1/N),
\end{aligned}
$$

where the last equality is because of the dominated convergence theorem. Consequently, we know that $\text{var}\big\{\big(Ne^{\alpha_N}\big)^{-1}$ $\ddot{\mathcal{L}}_{0k,j_1j_2}(\theta_k^*)\big\} \to 0$ when $N \to \infty$. Together with the result of $E\{(Ne^{\alpha_N})^{-1}\ddot{\mathcal{L}}_{0k}(\theta_k^*)\}$, we have proved that $\big(Ne^{\alpha_N}\big)^{-1}\ddot{\mathcal{L}}_{0k}(\theta_k^*) \to_p \Sigma_k^*$ as $N \to \infty$. This accomplishes the proof of (9).

### A.3. Proof of Theorem 2.3

In this appendix, we study the properties of the subsample-based pairwise maximum likelihood estimator $\widehat{\theta}_k^{\text{sub}}$ for $1 \leq k \leq K$. Specifically, we need to investigate a set of major–and–rare class pair logistic regression models as follows

$$
P\Big(Y_i = k \mid Z_i, Y_i = k \text{ or } Y_i = 0 \ \& \ b_i = 1\Big) = \frac{\exp(Z_i^\top\theta_k^* + \alpha_N^{\text{sub}})}{1 + \exp(Z_i^\top\theta_k^* + \alpha_N^{\text{sub}})},
$$

where $\alpha_N^{\text{sub}} = \alpha_N - \log\pi_N$. Define $w_{i,0k}^{\text{sub}} = P(Y_i = k|Z_i, Y_i = k \text{ or } Y_i = 0 \ \& \ b_i = 1)$. By the condition that $\pi_N/e^{\alpha_N} \to \infty$ as $N \to \infty$, we have $\alpha_N^{\text{sub}} \to -\infty$, which makes $w_{i,0k}^{\text{sub}} \to 0$ as $N \to \infty$. Consequently, the conditional log-likelihood function for the $k$–th two-class logistic regression problem can be written as $\mathcal{L}_{0k}^{\text{sub}}(\theta_k^*) = \sum_{i=1}^N a_{ik}\log w_{i,0k}^{\text{sub}} + a_{i0}b_i\log w_{i,00}^{\text{sub}} = \sum_{i=1}^N a_{ik}\big(Z_i^\top\theta_k^* + \alpha_N^{\text{sub}}\big) - \big(a_{ik} + a_{i0}b_i\big)\log\big\{1 + \exp\big(Z_i^\top\theta_k^* + \alpha_N^{\text{sub}}\big)\big\}$, where $\alpha_N^{\text{sub}} = \alpha_N - \log\pi_N$ and $P(b_i = 1) = \pi_N$ for some sampling probability $0 < \pi_N < 1$. Then a SPMLE can be obtained as $\widehat{\theta}_k^{\text{sub}} = \text{argmax}_{\theta_k^*}\mathcal{L}_{0k}^{\text{sub}}(\theta_k^*)$. Similarly with the proof of Theorem 2.1, to show that $\widehat{\theta}_k^{\text{sub}}$ is $\sqrt{Ne^{\alpha_N}}$–consistent and asymptotically normal, We would further prove in the following two steps that

$$
\big(Ne^{\alpha_N}\big)^{-1/2}\dot{\mathcal{L}}_{0k}^{\text{sub}}\big(\theta_k^*\big) \to_d N(0, \Sigma_k^*), \tag{11}
$$

$$
\big(Ne^{\alpha_N}\big)^{-1}\ddot{\mathcal{L}}_{0k}^{\text{sub}}\big(\theta_k^*\big) \to_p -\Sigma_k^*, \tag{12}
$$

as $N \to \infty$, where recall $\Sigma_k^* = E\big\{\exp(Z_i^\top\theta_k^*)Z_iZ_i^\top\big\}$. This finishes the proof of Theorem 2.3.

PROOF OF (11). We compute the mean and covariance of $\dot{\mathcal{L}}_{0k}^{\text{sub}}(\theta_k^*)$ respectively. Note that $\dot{\mathcal{L}}_{0k}^{\text{sub}}(\theta_k^*) = \sum_{i=1}^N\big\{a_{ik} - \big(a_{ik} + a_{i0}b_i\big)w_{i,0k}^{\text{sub}}\big\}Z_i$. Consequently, the mean of $\dot{\mathcal{L}}_{0k}^{\text{sub}}(\theta_k^*)$ can be calculate as $E\big\{\dot{\mathcal{L}}_{0k}^{\text{sub}}(\theta_k^*)\big\} = NE\big[\big\{a_{ik} - (a_{ik} + a_{i0}b_i)w_{i,0k}^{\text{sub}}\big\}Z_i\big] = NE\big[\big\{w_{ik}^* - (w_{ik}^* + w_{i0}^*\pi_N)w_{i,0k}^{\text{sub}}\big\}Z_i\big] = 0$ because of the fact that $(w_{ik}^* + w_{i0}^*\pi_N)w_{i,0k}^{\text{sub}} = w_{ik}^*$. Meanwhile, the covariance of $\dot{\mathcal{L}}_{0k}^{\text{sub}}(\theta_k^*)$ can be calculated as

$$
\begin{aligned}
&\text{cov}\Big\{\big(Ne^{\alpha_N}\big)^{-1/2}\dot{\mathcal{L}}_{0k}^{\text{sub}}\big(\theta_k^*\big)\Big\} = e^{-\alpha_N}E\Big[\big\{a_{ik} - \big(a_{ik} + a_{i0}b_i\big)w_{i,0k}^{\text{sub}}\big\}^2 Z_iZ_i^\top\Big] \\
=\ & e^{-\alpha_N}E\Big[\big\{a_{ik} + \big(a_{ik} + a_{i0}b_i\big)w_{i,0k}^{\text{sub}2} - 2a_{ik}w_{i,0k}^{\text{sub}}\big\}Z_iZ_i^\top\Big] \\
=\ & e^{-\alpha_N}E\Big[\big\{w_{ik}^* + (w_{ik}^* + w_{i0}^*\pi_N)w_{i,0k}^{\text{sub}2} - 2w_{ik}^* w_{i,0k}^{\text{sub}}\big\}Z_iZ_i^\top\Big] \\
=\ & e^{-\alpha_N}E\big\{w_{ik}^*(1 - w_{i,0k}^{\text{sub}})Z_iZ_i^\top\big\} = \Sigma_k^*\{1 + o(1)\},
\end{aligned}
$$

where the first equality is due to the fact that $E\big[\big\{a_{ik} - (a_{ik} + a_{i0}b_i)w_{i,0k}^{\text{sub}}\big\}Z_i\big] = 0$ and the third equality holds because $(w_{ik}^* + w_{i0}^*\pi_N)w_{i,0k}^{\text{sub}} = w_{ik}^*$.

We then verify the Lindeberg–Feller condition. For any $\epsilon > 0$,

$$
\sum_{i=1}^{N} E\left[\left\|\left\{a_{ik} - \left(a_{ik} + a_{i0}b_i\right)w_{i,0k}^{\mathrm{sub}}\right\}Z_i\right\|^2 I\left(\left\|\left\{a_{ik} - \left(a_{ik} + a_{i0}b_i\right)w_{i,0k}^{\mathrm{sub}}\right\}Z_i\right\| > \sqrt{Ne^{\alpha_N}}\epsilon\right)\right]
$$

$$
= NE\left\{w_{ik}^*(1 - w_{i,0k}^{\mathrm{sub}})^2\|Z_i\|^2 I\left(\left\|(1 - w_{i,0k}^{\mathrm{sub}})Z_i\right\| > \sqrt{Ne^{\alpha_N}}\epsilon\right)\right\}
$$

$$
+ N\pi_N E\left\{(1 - w_{ik}^*)w_{i,0k}^{\mathrm{sub}2}\|Z_i\|^2 I\left(\left\|w_{i,0k}^{\mathrm{sub}}Z_i\right\| > \sqrt{Ne^{\alpha_N}}\epsilon\right)\right\}
$$

$$
\leq NE\left\{w_{ik}^*\|Z_i\|^2 I\left(\|Z_i\| > \sqrt{Ne^{\alpha_N}}\epsilon\right)\right\} + N\pi_N E\left\{w_{i,0k}^{\mathrm{sub}2}\|Z_i\|^2 I\left(\left\|w_{i,0k}^{\mathrm{sub}}Z_i\right\| > \sqrt{Ne^{\alpha_N}}\epsilon\right)\right\}
$$

$$
\leq 2Ne^{\alpha_N}E\left\{e^{\|Z_i^\top\theta_k^*\|}\|Z_i\|^2 I\left(\|Z_i\| > \sqrt{Ne^{\alpha_N}}\epsilon\right)\right\} = o\left(Ne^{\alpha_N}\right).
$$

Combining the above results, we finish the proof of (11) by applying the Lindeberg–Feller central limit theorem (Van der Vaart, 2000).

PROOF OF (12). We compute the mean and covariance of $\ddot{\mathcal{L}}_{0k}^{\mathrm{sub}}(\theta_k^*)$ respectively. Note that $\ddot{\mathcal{L}}_{0k}^{\mathrm{sub}}(\theta_k^*) = -\sum_{i=1}^{N}\left(a_{ik} + a_{i0}b_i\right)w_{i,0k}^{\mathrm{sub}}(1 - w_{i,0k}^{\mathrm{sub}})Z_iZ_i^\top$. Hence, we obtain the mean of $\ddot{\mathcal{L}}_{0k}^{\mathrm{sub}}(\theta_k^*)$ as $E\{(Ne^{\alpha_N})^{-1}\ddot{\mathcal{L}}_{0k}^{\mathrm{sub}}(\theta_k^*)\} = -e^{-\alpha_N}E\{(a_{ik} + a_{i0}b_i)w_{i,0k}^{\mathrm{sub}}(1 - w_{i,0k}^{\mathrm{sub}})Z_iZ_i^\top\} = -e^{-\alpha_N}E\{(w_{ik}^* + w_{i0}^*\pi_N)w_{i,0k}^{\mathrm{sub}}(1 - w_{i,0k}^{\mathrm{sub}})Z_iZ_i^\top\} = -e^{-\alpha_N}E\{w_{ik}^*(1 - w_{i,0k}^{\mathrm{sub}})Z_iZ_i^\top\} = -\Sigma_k^*\{1 + o(1)\}$. We then compute the variance of the $(j_1, j_2)$–th element of $(Ne^{\alpha_N})^{-1}\ddot{\mathcal{L}}_{0k}^{\mathrm{sub}}(\theta_k^*)$ and have

$$
\mathrm{var}\left\{\left(Ne^{\alpha_N}\right)^{-1}\ddot{\mathcal{L}}_{0k,j_1j_2}^{\mathrm{sub}}(\theta_k^*)\right\} = N^{-1}e^{-2\alpha_N}E\left\{\left(a_{ik} + a_{i0}b_i\right)w_{i,0k}^{\mathrm{sub}2}\left(1 - w_{i,0k}^{\mathrm{sub}}\right)^2 Z_{ij_1}^2 Z_{ij_2}^2\right\}
$$

$$
- N^{-1}e^{-2\alpha_N}E^2\left\{\left(a_{ik} + a_{i0}b_i\right)w_{i,0k}^{\mathrm{sub}}\left(1 - w_{i,0k}^{\mathrm{sub}}\right)Z_{ij_1}Z_{ij_2}\right\}
$$

$$
= N^{-1}e^{-2\alpha_N}\left[E\left\{w_{ik}^* w_{i,0k}^{\mathrm{sub}}\left(1 - w_{i,0k}^{\mathrm{sub}}\right)^2 Z_{ij_1}^2 Z_{ij_2}^2\right\} - E^2\left\{w_{ik}^*\left(1 - w_{i,0k}^{\mathrm{sub}}\right)Z_{ij_1}Z_{ij_2}\right\}\right]
$$

$$
= N^{-1}\mathrm{var}\left\{\exp\left(Z_i^\top\theta_k^*\right)Z_{ij_1}Z_{ij_2}\right\}\{1 + o(1)\} = O(1/N),
$$

where the last equality is because of the condition that $w_{i,0k}^{\mathrm{sub}} \to 0$ as $N \to \infty$ and the dominated convergence theorem. Hence, we know that $\mathrm{var}\{(Ne^{\alpha_N})^{-1}\ddot{\mathcal{L}}_{0k,j_1j_2}^{\mathrm{sub}}(\theta_k^*)\} \to 0$ as $N \to \infty$. Together with the result of $E\{(Ne^{\alpha_N})^{-1}\ddot{\mathcal{L}}_{0k}^{\mathrm{sub}}(\theta_k^*)\}$, we have proved that $(Ne^{\alpha_N})^{-1}\ddot{\mathcal{L}}_{0k}^{\mathrm{sub}}(\theta_k^*) \to_p \Sigma_k^*$ as $N \to \infty$. This finishes the proof of (12).

### A.4. Illustration Details

In our TTS dataset analysis, there are total of $N$=56,016 sub-images belonging to nine classes, where the first class (i.e., $k = 0$) represents the background class; the remainder are car model classes. The major class (i.e., $k = 0$) accounts for more than 95% of the total sample. The percentages of other classes are bar-plotted in Figure 6. As shown, this dataset aligns well with our theoretical framework.

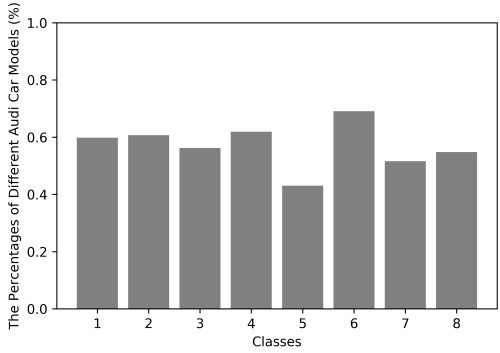

*Figure 6.* Sample percentages of different Audi car models.

