# OpenReview forum: "Pairwise Maximum Likelihood For Multi-Class Logistic Regression Model With Multiple Rare Classes"
_ICML.cc/2025/Conference — ICML 2025 poster_

### Official Review · Reviewer_rHhH · 2025-02-26

**Overall Recommendation:** 4

**Summary:**

This paper addresses the problem of multi-class logistic regression in scenarios with class imbalance: specifically, one class is overwhelmingly dominant and the remaining classes are rare. The authors develop a theoretical framework demonstrating that, under appropriate assumptions and asymptotic conditions, the maximum likelihood estimator (MLE) is asymptotically normal with a mean-zero error, and has an asymptotically block-diagonal covariance structure, implying that the parameters for the rare classes are asymptotically independent. Building on this insight, the paper proposes a pairwise maximum likelihood estimator (PMLE) that decomposes the multi-class problem into separate binary logistic regression problems, along with a subsample-based variant (SPMLE) that reduces computational costs by down-sampling the major class.

## Update After Rebuttal
I raised my score as a result of a discussion with the authors (see below).

**Claims And Evidence:**

### Interesting theoretical results with limited motivation.
The paper presents intriguing theoretical results that decompose a multi-class classification problem with imbalanced classes into a set of pairwise binary problems. However, the overall motivation and practical impact remain questionable. The introduction discusses customer purchase behavior and related contexts, which might lead readers to expect approaches involving gaze tracking or attention mechanisms integrated into an object detection pipeline. Instead, the paper ultimately focuses on logistic regression applied to sub-image classification. For instance, one might naturally consider a two-stage approach—first detecting car plates and then classifying them—to mitigate the imbalance issue, an alternative not discussed by the authors.

### Strong assumptions with limited practical relevance.
The theoretical guarantees rely on specific conditions such as $\alpha_N \to -\infty$ and $a_N + \log N \to \infty$. These conditions ensure that while the probability of a rare event converges to zero, the absolute number of rare events still diverges, allowing for asymptotic analysis. In practice, however, these assumptions may not hold. For example, in a dataset of car images, the probability that a car plate is visible might remain roughly constant regardless of $N$ (assuming each image reliably shows a car plate). Moreover, the convergence rate is effectively downgraded from $1/\sqrt{N} \to 1/\sqrt{N e^{a_N}}, meaning that when the rare events are very infrequent (i.e., $e^{a_N}$ is very small), the effective sample size for estimation is substantially reduced, resulting in a much slower convergence. This raises concerns about the practical feasibility of the method even if the theoretical assumptions are met.

### Limited experimental validation.
While the paper supports its theoretical claims with simulation studies and a single real-world TikTok Screenshots dataset, the experimental evidence is limited. The evaluation compares only the authors’ own MLE-based variants (GMLE, PMLE, and SPMLE) without benchmarking against alternative methods—such as a detection-classification pipeline or other imbalance-handling techniques (e.g., cost-sensitive learning, upsampling, or downsampling) —that are commonly used in practice. This narrow scope of experiments makes it challenging to assess the broader impact and practical relevance of the proposed approach.

Overall, although the paper’s theoretical contributions are solid, the motivation, practical relevance, and experimental validation leave several claims insufficiently supported by convincing evidence.

**Essential References Not Discussed:**

The paper would benefit from a broader discussion of the imbalanced classification literature. For example, recent advances in deep learning for imbalanced data, such as focal loss (Lin et al., ICCV 2017) and class-balanced loss based on the effective number of samples (Cui et al., CVPR 2019), have shown promising results in mitigating imbalance issues, especially in object detection tasks. Discussing these methods would help situate the paper’s contributions within the broader context of imbalanced classification and highlight potential avenues for extending the theoretical results to more complex models.

**Experimental Designs Or Analyses:**

I incorporated experimental concerns into the above “Claims and Evidence” section.

**Methods And Evaluation Criteria:**

I incorporated methodological concerns into the above “Claims and Evidence” section.

**Other Comments Or Suggestions:**

The paper was an enjoyable read. Please note that my comments represent my initial impressions and may include misunderstandings. I welcome further discussion on these points and am open to revising my score once my questions and concerns are adequately addressed.

**Other Strengths And Weaknesses:**

### Other strengths to note:
Building on their theoretical insights, the authors introduce the pairwise maximum likelihood estimator PMLE and its subsample-based variant, SPMLE. SPMLE keeps the same convergence rate as PMLE, efficiently reduce computational burdens without sacrificing benefits.

### Minor weaknesses:
- The term “NR algorithm” is undefined. I assume it refers to the Newton-Raphson algorithm, but a clear definition or brief explanation would improve clarity.
- To ensure reproducibility, it would be beneficial if the authors made the code available.

**Questions For Authors:**

I have incorporated questions into the above sections.

**Relation To Broader Scientific Literature:**

Their theoretical findings, notably the asymptotically block-diagonal covariance structure and the derivation of a convergence rate, provide rigorous insights into the behavior of maximum likelihood estimators under extreme imbalance. While the focus is on logistic regression, these insights have the potential to inform future research on more complex models, advancing our understanding of algorithm design in extreme imbalance scenarios.

**Theoretical Claims:**

I have reviewed the high-level theoretical claims but I did not perform a line-by-line verification of the detailed proofs in the main text and Appendix. Consequently, while the arguments appear plausible, there remains a possibility that oversights exist.

---

> ### Author Rebuttal · Authors · 2025-03-31
>
> We thank the reviewer for the constructive suggestions. The concerns have been well addressed as follows.
>
> 1. **Motivation and Practical Impact.**
>     - **Theoretical Motivation.** The focus here is the theoretical investigation of logistic regression with rare classes. Specifically, we find that the asymptotic covariance of the resulting MLE is block-diagonal, which further inspires the novel PMLE method. All the discussions in the introduction about customer purchase behavior are used to demonstrate the practical importance of our problem. However, with your kind reminder, we have realized that we might have overemphasized this point. We now have rewritten the section for better elaboration.
>     - **Comparison with Two-Stage Approach.** We consider a very ideal situation with all car plates having been correctly detected. All rare classes have been perfectly separated from the major one. We then only need to focus on rare classes. The detailed classification results on the TikTok Screenshots (TTS) dataset are given in Table A.1. Even under this ideal situation, the resulting classification accuracy (RARE) is much worse than our one-stage strategy (GMLE, PMLE and SPMLE). This is not surprising since the useful information contained in the major class was not effectively used in the second stage. In contrast, all the information from both major and rare classes is comprehensively used by the one-stage strategies.
>
> ### Table A.1: Prediction results for the TTS dataset.
> |  | GMLE | PMLE | SPMLE | RARE |
> | --- | --- | --- | --- | --- |
> | ACC | 0.836 | 0.835 | 0.824 | 0.745 |
> | AUC | 0.997 | 0.999 | 0.999 | 0.969 |
>
> 2. **Theoretical Properties.**
>     - **Assumption.** The theoretical assumption $\alpha_N \to -\infty$ and $\alpha_N+\log N \to \infty$ are used for allowing rigorous asymptotic analysis. This leads to the block diagonal structure of the asymptotic covariance for MLE, and further leads to the novel PMLE method. Without this specific condition, we can never obtain those theoretical findings and then be inspired to develop PMLE method. Moreover, those conditions seem to be very standard in the literature. See, for example, Equation (2) in Wang (2020, ICML) and Section 2 in Wang et al. (2021, NeurIPS).
>     - **Convergence Rate.** The convergence rate reflects that sufficient amount of samples must be provided for both major and every rare class. Otherwise, no consistent parameter estimates can be obtained. The much slower convergence rate of $1/\sqrt{Ne^{\alpha_N}}$ is as expected. In fact, this is a phenomenon having been well documented in the literature. See, for example, Wang (2020, ICML), Song and Zou (2024, TIT), and Wang et al. (2021, NeurIPS). We now have made this point clear in the revision.
>
> 3. **Empirical Comparison with Baseline Methods.** We have included the following methods for comparison on the TTS dataset: the focal loss (FL) of Lin et al. (2017), the class-balanced loss (CBL) of Cui et al. (2019), the cost sensitive loss (CSL) and random downsampling (RDS) of Fernández et al. (2018). All methods are optimized according to the suggestions of the original papers. The results are summarized below in Table A.2. We are happy to report that our method outperforms all competitors.
>
> ### Table A.2: Prediction results for the TTS dataset.
> |  | GMLE | PMLE | SPMLE | FL | CBL | CSL | RDS |
> | --- | --- | --- | --- | --- | --- | --- | --- |
> | ACC | 0.836 | 0.835 | 0.824 | 0.794 | 0.789 | 0.747 | 0.763 |
> | AUC | 0.997 | 0.999 | 0.999 | 0.998 | 0.998 | 0.991 | 0.996 |
>
> 4. **Minor Issues.** For the other issues, we have also addressed them carefully.
>     - “NR algorithm” has been referred to as the Newton-Raphson algorithm in the revision.
>     - The code has been made publicly available at [https://anonymous.4open.science/r/Anony-63CC](https://anonymous.4open.science/r/Anony-63CC).
>
> **Additional References**
> 1. Cui, Y., Jia, M., Lin, T. Y., Song, Y., & Belongie, S., 2019, Class-balanced loss based on effective number of samples. CVPR.
> 2. Lin, T. Y., Goyal, P., Girshick, R., He, K., & Dollár, P., 2017, Focal loss for dense object detection. ICCV.
> 3. Song, Y., & Zou, H., 2024, Minimax optimal rates with heavily imbalanced binary data. IEEE TIT.
> 4. Fernández, A., García, S., Galar, M., Prati, R. C., Krawczyk, B., & Herrera, F, 2018, Learning from imbalanced data sets. Springer.

---

> > ### Comment · Reviewer_rHhH · 2025-04-03
> >
> > I greatly appreciate the authors’ response in thoroughly clarifying my concerns. I have raised my score accordingly.

---

> > > ### Author Response · Authors · 2025-04-03
> > >
> > > We truly appreciate the valuable time and effort you have dedicated to reviewing our submission. Thank you so much for your support.

---

### Official Review · Reviewer_wb9a · 2025-03-15

**Overall Recommendation:** 4

**Summary:**

This paper focuses on multi-class logistic regression with one major class and multiple rare classes, a problem arising from real applications.
By the suggestions from Theorem 2.1, the standard maximum likelihood estimators as well as the re-parametrized version are asymptotically independent for different rare classes, which in turn motivates the development of a new algorithm,  PMLE, by solving multiple pairwise log-likelihood functions. Since the pairwise log-likelihood functions contain the samples with large size in the major class,  minimizing the pairwise log-likelihood functions is still computationally challenging. To further accelerate the computation, the authors propose to solve subsample-based pairwise log-likelihood functions, termed as SPMLE. Theoretically, the authors prove that the newly-proposed algorithms, PMLE and SPMLE, require lower computational cost compared to the standard maximum likelihood estimators without compromising asymptotic efficiency.

**Claims And Evidence:**

Yes.

**Essential References Not Discussed:**

No

**Experimental Designs Or Analyses:**

Yes. The proposed methods are corroborated by simulation studies and real-example analysis.

**Methods And Evaluation Criteria:**

Yes.

**Other Comments Or Suggestions:**

Typographical remarks:

1. In Line 93, it should be $1\le i\le N$.

**Other Strengths And Weaknesses:**

**Strengths:**

1. Inspired by the asymptotic covariance matrix of the global MLE,  the manuscript  proposed two methods, PMLE and SPMLE,  which
have a significant advantage in computation and are specifically designed for the problem of multi-class logistic regression with one major class and multiple rare classes.

2. The established theory for PMLE and SPMLE  is quite impressive.  It reveals that the newly-proposed algorithms require lower computational cost without compromising asymptotic efficiency.

3. Comprehensive numerical studies, including simulations and real-data analyses, demonstrate the effectiveness of the proposed methods.


Weaknesses:

The appendix should be well-organized. For instance, Lines 629-639 currently appear cluttered and could be improved by using interline formulas for better clarity and readability.

**Questions For Authors:**

In this work, the number of classes $K$ is treated as fixed. I think it may be interesting to investigate the theoretical behavior of the proposed method when considering the case where  $K$ diverges as the number of samples increases?

**Relation To Broader Scientific Literature:**

A result similar to Theorem 2.1 in the submitted manuscript is also established by Wang (2020); however, their work focuses on an imbalanced two-class problem, whereas the manuscript addresses a multi-class setting with one major class and multiple rare classes. Notably, a new computationally efficient algorithm with theoretical guarantee is proposed in the manuscript, advancing beyond the scope of prior work.

[1] Wang, H. Logistic regression for massive data with rare events. In International Conference on Machine Learning, pp. 9829–9836. PMLR, 2020.

**Theoretical Claims:**

Yes.

---

> ### Author Rebuttal · Authors · 2025-03-31
>
> We thank the reviewer for all the constructive suggestions. All the concerns have been well received and carefully addressed as follows.
>
> 1. **Diverging Number of Classes.**
> With a diverging $K$, the expected percentage of rare classes should be even smaller.  To ensure a diverging sample size for each rare class, the technical assumption $\alpha_N \to -\infty$ should be replaced by $\alpha_N + \log K \to -\infty$. The resulting theoretical behavior of our proposed methods remains fairly the same. More specifically, the convergence rate of the PMLE remains $\sqrt{Ne^{\alpha_N}}$-consistent. The PMLEs associated with different rare classes remain mutually independent asymptotically. More importantly, the resulting PMLE should be asymptotically as efficient as GMLE. To numerically demonstrate this point, we replicate the simulation example but with $K = [N^{0.25}]$. See below Table A for the detailed simulation results. We find that (1) both GMLE and PMLE remain statistically consistent, and (2) their asymptotic efficiency seems to be the same with extremely similar RMSE values. We now have made this point clear in the revision.
>
> ### Table A: Simulation results with diverging $K$.
> | N         | $10^4$ | $2.5\times10^4$ | $5\times10^4$ | $7.5\times10^4$ | $10^5$ |
> |-----------|--------|----------------|--------------|-----------|--------|
> | K         | 10     | 12             | 14           | 16             | 17     |
> | GMLE      | 0.114  | 0.091          | 0.077        | 0.068          | 0.064  |
> | PMLE      | 0.114  | 0.091          | 0.077        | 0.068          | 0.064  |
> | SPMLE     | 0.116  | 0.093          | 0.078        | 0.069          | 0.064  |
>
> 2. **Minor Issues.** For the other issues that you have mentioned, we have also corrected them carefully.
>
> (1) In the revision, we have reorganized Lines 629-639 in the Appendix.
>
> (2)  In Line 93, it should be $1 \le i \le N$. We now have made it clear in the revision.

---

> > ### Comment · Reviewer_wb9a · 2025-04-03
> >
> > Thank you for your detailed response! I will keep my already positive score.

---

> > > ### Author Response · Authors · 2025-04-05
> > >
> > > Thank you for the valuable time and effort you have spent reviewing our submission. We are truly grateful for your support of our work.

---

### Official Review · Reviewer_6Ats · 2025-03-16

**Overall Recommendation:** 3

**Summary:**

This paper studies the parameter estimation problem for Multi-class logistic model with one major class and $K$ rare classes. The main observation is that, under certain decay rate assumptions on the coefficients of rare classes. The joint MLE estimator is asymptotically equal to the pairwise MLE estimator between each rare class and major class, and the pairwise MLE estimator significantly save the computation complexity. Authors also extends the theoretical guarantee of pairwise estimators to the sub-sampled setting.

**Claims And Evidence:**

The theoretical statements and contributions made in this paper are easy to understand. However, there are several points that could further improve readability:


1. I am a bit confused about the estimator under the model described in Equation (1). From my understanding, all subsequent operations (e.g., pairwise estimates, sub-sampling) are based on the model defined in Equation (2). Why not start directly with the model in Equation (2)? Additionally, the reductions made from the model in Equation (1) to the model in Equation (2) (paragraph 2 in section 2.2) should be explained more clearly.

2. Please consider formally stating the technical assumptions mentioned in the last paragraph of Section 2.1.

**Essential References Not Discussed:**

No

**Experimental Designs Or Analyses:**

No

**Methods And Evaluation Criteria:**

The proposed method(pairwise MLE) is the main contribution of this paper, and is easy to apply. The applied evaluation criteria(estimation error) is standard.

**Other Comments Or Suggestions:**

No

**Other Strengths And Weaknesses:**

Strength:

The studied setting is new, and the proposed method is new and efficient.

Weakness:

1. In my opinion, the assumptions made regarding rare classes are too strong, which weakens the theoretical contributions of this paper. Specifically, the theoretical guarantees require the rare classes to be sufficiently balanced and impose an explicit rate bound on $\alpha_N$. I tend to believe this is a relatively loose sufficient condition, as no lower bound results or counterexamples are provided to justify its necessity.

2. The assumption that there exists only one major class (corresponding to $k=0$) seems restrictive. Can the current results be extended to a more general setting where there are $m$ major classes?

**Questions For Authors:**

My questions are:

1. Are there any necessity results on the decay rate condition $\alpha + \log N \to +\infty$ ?

2. Can the assumption of a single major class be extended to accommodate a general case with $m$ major classes?

**Relation To Broader Scientific Literature:**

This paper studies a special structure in multi-class logistic regression, which is new in this area. And the proposed new estimator provides a efficient algorithm under such structure, as demonstrated in their experiment.

**Theoretical Claims:**

I have not checked all the details of the theoretical proofs, but the independence structure revealed by Theorem 2.1 make sense to me given the assumptions made in this paper. And the Theorem 2.2 and 2.3 also seems reasonable given such independence structure.

---

> ### Author Rebuttal · Authors · 2025-03-31
>
> We thank the reviewer for all the constructive suggestions. All the concerns have been well received and carefully addressed as follows.
>
> 1. **Relationship between Two Models.** Note that the model (2) is a special case of the model (1), which is a more general model. As you have correctly noted, we can indeed start with the model (2) directly. However, we then lose the opportunity to explain why the model (2) is specified in its current form. Specifically, by starting from the general model (1), it seems clear that we have to set $\alpha_{N k}\rightarrow -\infty$. Otherwise, the rare class phenomenon cannot be theoretically reflected. Moreover, we cannot have balanced rare classes unless $\alpha_{N k_1}-\alpha_{N k_2}=O(1)$ for any $k_1\neq k_2$. Both the conditions (i.e., $\alpha_{N k}\rightarrow -\infty$ and $\alpha_{N k_1}-\alpha_{N k_2}=O(1)$) enable us to re-parameterize $\alpha_{Nk} = \alpha_{N}+\beta_{0k}$. That leads to the model (2). We now have explained this issue in the revision more clearly. Thank you so much for this kind reminder.
>
> 2. **Imbalanced Class Assumption.**
> Requiring the rare classes to be sufficiently balanced is a relatively loose sufficient condition and not very necessary. In fact, if rare classes are unbalanced, PMLE can still be applied without any difficulty.  However, there is one critical condition.
> That is the sample size of the target rare class must be sufficiently large. Otherwise, no consistent classifier can be learned. This seems to be a very basic and reasonable condition, which has been widely used in the past literature (Wang, 2020, ICML; Li et al., 2024, JMLR; Wang et al., 2021, NeurIPS). However, this very basic and reasonable condition can be easily violated, if rare classes are highly unbalanced. In such cases, some classes may be even rarer than others. We refer to those even rarer classes as tiny classes, whose sample sizes are often too tiny to support any meaningful statistical learning. This is the only reason we focus on the balanced case in our theoretical analysis. We now have made it clear in the revision.
>
> 3. **Multiple Major Classes.** Our method can be readily extended to a more general setting with multiple major classes. To fix the idea, consider, for example, the case with two major classes. Denote the two major classes by $k = 0$ and $k = 1$, respectively. Then, the logistic regression model becomes:
> $$
> P(Y\_i = 0 \mid X\_i)=\frac{1}{1+\exp(Z\_i^{\top} \theta^*\_1)+\sum\_{k = 2}^K \exp(Z\_i^{\top} \theta^*\_k+\alpha\_N)}, $$
> $$P(Y\_i = 1 \mid X\_i)=\frac{\exp(Z\_i^{\top} \theta^*\_1)}{1+\exp(Z\_i^{\top} \theta^*\_1)+\sum\_{k = 2}^K \exp(Z\_i^{\top} \theta^*\_k+\alpha\_N)}, (A.1)$$
> $$P(Y_i = k \mid X\_i)=\frac{\exp(Z\_i^{\top} \theta^*\_k+\alpha_N)}{1+\exp(Z\_i^{\top} \theta^*\_1)+\sum_{k = 2}^K \exp(Z\_i^{\top} \theta^*\_k+\alpha\_N)} \text{ for }2 \leq k \leq K. (A.2)$$
> The key difference between (A.1) and (A.2) is that the diverging intercept $\alpha_N$ is involved in (A.2) for rare classes but not in (A.1) for the major class. To estimate the model parameters, the pairwise log-likelihood in Section 2.4 remains valid. The only difference is that the convergence rate of $\hat\beta_1$ becomes $\sqrt{N}$. However, the convergence rate of the parameters associated with the rare classes (i.e., $\hat\beta_k$ for $2 \leq k \leq K$) remains $\sqrt{N e^{\alpha_N}}$. We now have made it clear in the revision.
>
> 4. **Minor issues.** For the other issues that you have mentioned, we have also corrected them carefully.
>
> (1) We have now formally stated the technical assumptions in the last paragraph of Section 2.1 as: "Assume (C1) $\alpha_{N k}\rightarrow -\infty$ and (C2) $\alpha_{N k}+\log N\rightarrow \infty$ as $N\rightarrow \infty$ for every $1\leq k\leq K$."
>
> (2) We have carefully discussed the decay rate condition $\alpha_N+\log N\rightarrow \infty$. Without this condition, we might have $\alpha_N + \log N \to -\infty$. Then, the expected sample size for the $k$th rare class becomes $E(N_k) \approx NP(Y_i = k) \to 0$ as $N \to \infty$ for every $1\leq k \leq K$. As a consequence, the sample sizes for the rare classes may be too small to yield any consistent estimator.
>
> **Additional Reference**
>
> Wang, H., Zhang, A., & Wang, C., 2021, Nonuniform negative sampling and log odds correction with rare events data. NeurIPS.

---

### Decision · Program_Chairs · 2025-05-01

**Decision:**

Accept (poster)

**Comment:**

The paper combines theoretical investigations and algorithmic results inspired by real-world problems related to multi-class classification. Reviewers appreciated the novelty and efficiency of the proposed methods, specifically designed to solve multi-class classification tasks in case of strong class imbalance. The theoretical analysis of the problem is based on the study of the statistical properties of the MLE and requires some assumptions that raised some concerns about the practical relevance of the results; nevertheless, the effectiveness of the proposed strategy has been appreciated by all referees and therefore I will therefore recommend *Accept*.